# Genome-wide cross-disease analyses highlight causality and shared biological pathways of type 2 diabetes with gastrointestinal disorders

Emmanuel O. Adewuyi [1,2] ✉, Tenielle Porter [1,2,3], Eleanor K. O'Brien [1,2], Oladapo Olaniru [4], Giuseppe Verdile[3,5] & Simon M. Laws [1,2,3] ✉

Studies suggest links between diabetes and gastrointestinal (GI) traits; however, their underlying biological mechanisms remain unclear. Here, we comprehensively assess the genetic relationship between type 2 diabetes (T2D) and GI disorders. Our study demonstrates a significant positive global genetic correlation of T2D with peptic ulcer disease (PUD), irritable bowel syndrome (IBS), gastritis-duodenitis, gastroesophageal reflux disease (GERD), and diverticular disease, but not inflammatory bowel disease (IBD). We identify several positive local genetic correlations (negative for T2D – IBD) contributing to T2D's relationship with GI disorders. Univariable and multivariable Mendelian randomisation analyses suggest causal effects of T2D on PUD and gastritis-duodenitis and bidirectionally with GERD. Gene-based analyses reveal a gene-level genetic overlap between T2D and GI disorders and identify several shared genes reaching genome-wide significance. Pathway-based study implicates leptin (T2D – IBD), thyroid, interferon, and notch signalling (T2D – IBS), abnormal circulating calcium (T2D – PUD), cardiovascular, viral, proinflammatory and (auto)immune-mediated mechanisms in T2D and GI disorders. These findings support a risk-increasing genetic overlap between T2D and GI disorders (except IBD), implicate shared biological pathways with putative causality for certain T2D – GI pairs, and identify targets for further investigation.

Type 2 diabetes (T2D) is a chronic metabolic disorder characterised by impaired response of insulin-sensitive tissues to insulin (insulin resistance), leading to hyperglycaemia and, ultimately, insulin secretion deficits[1,2]. Globally, the prevalence of diabetes has reached a pandemic proportion, with a recent estimate ranking it as 'one of the fastest growing health emergencies of the 21st century'[3]. In 2021, over 536 million people lived with diabetes worldwide, resulting in more than 6.7 million mortalities, with T2D alone accounting for 90–95% of all cases[3]. Unlike type 1 diabetes, which is aetiologically differentiated by autoimmune destruction of pancreatic beta cells, T2D is marked by chronic inflammation, insulin resistance and subsequent hyperinsulinemia, hyperglycaemia, and (in chronic stages) inadequate compensation for insulin secretory mechanisms[1,2]. The exact aetiology and underlying biology of T2D are still not fully understood.

However, a clear line of evidence supports its multifaceted and multifactorial pathogenesis, including genetics and lifestyle factors[4].

An important observation in T2D is that multiple organs and systems, particularly the gastrointestinal (GI) system, are involved or linked with the disorder's pathobiology[5,6]. For instance, the core pathological basis of T2D, failure of the pancreatic beta cells, relates to insulin resistance in the liver and the skeletal muscles—implicating the GI and musculoskeletal systems in the disorder's pathophysiology[2,7]. The GI system is central to glucose homeostasis[6], and approximately 75% of people with diabetes experience GI symptoms ranging from dysphagia, nausea, heartburn, bloating, and diarrhoea to abdominal pain and constipation[8–10]. Further supporting a potential involvement of GI traits in T2D, gut microbiota dysbiosis implicated in GI disorders[11,12] and *Helicobacter pylori* infection (a known risk

[1]Centre for Precision Health, Edith Cowan University, Joondalup 6027 Western, Australia. [2]Collaborative Genomics and Translation Group, School of Medical and Health Sciences, Edith Cowan University, Joondalup 6027 Western, Australia. [3]Curtin Medical School, Curtin University, Bentley 6102 Western, Australia. [4]Department of Diabetes, School of Cardiovascular and Metabolic Medicine & Sciences, King's College London, London, UK. [5]Curtin Health Innovation Research Institute, Curtin University, Bentley 6102 Western, Australia. ✉e-mail: e.adewuyi@ecu.edu.au; s.laws@ecu.edu.au

factor for peptic ulcer disease, PUD) have been suggested to contribute to a greater risk of T2D[13–17].

Notably, in several observational studies, T2D is associated with an increased risk of GI disorders, including gastroesophageal reflux disease (GERD)[18,19], PUD[20] (and its complication: peptic ulcer bleeding (PUB)[21,22]), and inflammatory bowel disease (IBD)[5,20,23,24]. In a nationwide Taiwanese population-based study, for example, individuals with T2D (cases) had a higher cumulative PUB hazard than controls [$P < 0.001$][21]. Following adjustment for other predictor variables, diabetes remained significantly associated with an increased risk of PUB (hazard ratio: 1.44, $P < 0.001$)[21]. The results from a systematic review and meta-analysis reinforced this finding, revealing a 43.3% (odds ratio [OR]: 1.43, 95% CI: 1.28–1.60) and 44.2% (OR: 1.44, 95% CI: 1.25–1.66) increase in PUB morbidity and 30-day mortality risk, respectively, in individuals with diabetes compared to controls[22]. A significant association of diabetes with GERD (OR: 1.61; $P = 0.003$) was also reported in another systematic review and meta-analysis[18]. Similarly, population-based studies have demonstrated an increased risk of T2D and prediabetes among individuals with IBD[23,24] and irritable bowel syndrome (IBS)[25], respectively.

Despite growing evidence for T2D's relationship or co-occurrence with GI disorders[5,8–10,18–25], biological mechanisms underlying the disorders and their potential comorbid relationships remain largely unresolved. For instance, it is unclear whether the increased incidence of GI disorders with T2D is due to the direct pathogenic effects of T2D or vice versa. While shared risk factors and common genetic aetiology may contribute, data in this respect are limited, especially regarding T2D's association with common GI disorders, such as gastritis, duodenitis, IBS, PUD, and diverticular disease. Regardless of their poorly understood mechanisms, consistent reports linking GI conditions with diabetic complications and poor glycaemic control[26,27] suggest that comorbidity of T2D with GI disorders represents a considerable source of morbidity and mortality in affected individuals.

Co-occurrence of T2D with GI disorders can, for instance, accelerate disease progression and contribute to increased healthcare costs, with the potential for wide-ranging adverse outcomes, including complicated management plans or worsened quality of life in individuals affected[28,29]. On the other hand, T2D's comorbid association with GI disorders may be involved in the disorders' pathogenesis through shared genetic aetiology, causal relationships, or common biological pathways. Thus, understanding the relationship between T2D and GI disorders can advance knowledge of their underlying biological mechanisms, identify targets for further investigation, and provide opportunities for treatment development, evidence-based clinical decisions, or precision prevention strategies.

So far, evidence of the potential relationship of T2D with GI disorders comes primarily from conventional observation studies[5,18–25]. Limitations such as reverse causation, measurement errors, and residual confounding often complicate the traditional observational study approach. An unbiased genetic method based on the analysis of world-leading genetic data is less susceptible to these limitations and, thus, provides a robust avenue for testing causal relationships and shared genetic susceptibility of one trait with another[30–32]. A few studies have assessed the potential causality of diabetes with GERD[33], diverticular disease[34], and some GI traits[35]. However, several aspects of the association between T2D and GI disorders remain understudied, and our knowledge of their underlying biology and potential comorbid relationships is incomplete. Beyond assessing causal associations, the present study provides new insights (to the best of our knowledge) into this subject through a more comprehensive but targeted focus on T2D and six common GI disorders—GERD, PUD, IBS, gastritis-duodenitis, diverticular disease, and IBD.

Briefly, PUD may be defined as a mucosal break (ulcer) larger than 3–5 mm in the stomach (gastric ulcer) or the upper small intestine (duodenal ulcer), resulting in abdominal pain and bloating[36]. Similar to PUD, gastritis-duodenitis is the inflammation of the stomach lining (gastritis) and the first part of the small intestine (duodenitis), often associated with abdominal pain and discomfort. IBS is a persistent disorder affecting bowel function, presenting as abdominal pain or discomfort linked to changes in bowel habits[37]. GERD commonly occurs when stomach contents (acid or nonacidic reflux) flow back into the oesophagus, leading to potential erosion and causing heartburn and upper abdominal pain[38]. Diverticulosis is characterised by sac-like protrusions (diverticula) in the colon, which may remain asymptomatic but can also lead to inflammation (diverticulitis) and colonic bleeding[39]. Diverticular disease is when diverticulosis becomes symptomatic and is typically marked by abdominal pain, bloating, and alterations in bowel habits[39]. IBD encompasses Crohn's disease and ulcerative colitis, characterised by chronic inflammation of the GI tract and associated symptoms such as abdominal pain, diarrhoea, and bloody stools[40].

Leveraging several large-scale genome-wide association studies (GWAS) summary statistics, this study addresses the question of genetic overlap by estimating global and local genetic correlations between T2D and the named common GI disorders. We also assess gene-level genetic overlap, estimate causality, and subsequently identify genes and biological pathways shared by T2D and GI disorders, providing insights into their underlying biological mechanisms. Our findings reveal a significant positive global genetic correlation between T2D and GI disorders, including PUD, IBS, gastritis-duodenitis, GERD, diverticular disease, and medicated phenotype for GERD/PUD (PGM) but not IBD. We demonstrate several positive local genetic correlations (negative for T2D-IBD) between T2D and GI disorders. Mendelian randomisation suggests that genetic liability to T2D has causal effects on PUD and gastritis-duodenitis and a bidirectional causal association with GERD. Moreover, gene-based association analysis reveals gene-level genetic overlap between T2D and GI disorders and identifies several shared genes reaching genome-wide significance. Our pathway-based study implicates leptin (T2D-IBD), thyroid, interferon, and notch signalling (T2D-IBS), QT interval anomaly, and abnormal circulating calcium (T2D-PUD), viral, proinflammatory, and (auto)immune-mediated mechanisms in T2D and GI disorders. Thus, the present study offers comprehensive and systematic evidence of the genetic relationship between T2D and GI disorders and identifies targets for follow-up investigations.

## Results

This study comprises five broad components (Fig. 1). We first utilised the 'linkage disequilibrium score regression analysis' method (LDSC)[41] to estimate global (genome-wide) SNP-based genetic correlation between T2D and six GI disorders, including PUD, IBS, gastritis-duodenitis, IBD, GERD, and diverticular disease. Second, in addition to understanding the average genetic effect represented by the genome-wide correlation (using LDSC), we utilised LAVA (Local Analysis of [co] Variant Association)[42] to identify specific genomic effects and regions contributing to the relationship between T2D and GI disorders. Third, we used bidirectional univariable and multivariable Mendelian randomisation (MR) analysis to evaluate and estimate the potential causal association between T2D and GI disorders. Fourth, we conducted gene-based analyses and used the results to assess whether more genes overlap between T2D and GI disorders than would be expected by chance.

Finally, we identified genes and biological pathways shared by T2D and GI disorders to gain mechanistic insights into their underlying biology. The present study used GWAS summary data for T2D and GI disorders sourced from various research consortia and public repositories. Additionally, we used a newly developed, medicated phenotype for GERD/PUD (PGM)[43] for analysis in global genetic correlation assessment. This phenotype was formulated by combining disease diagnoses for GERD or PUD with the respective use of medications for each condition[43]. The combination follows the acid-related nature of both GERD and PUD, with medicines for PUD known to exhibit therapeutic effects on GERD in clinical practice. Comprehensive information on this phenotype has been published[43]. Figure 1 provides a workflow summarising the methods of analysis implemented in this study. Table 1 summarises the data used for analysis, while Supplementary Data 1 provides additional and more specific information.

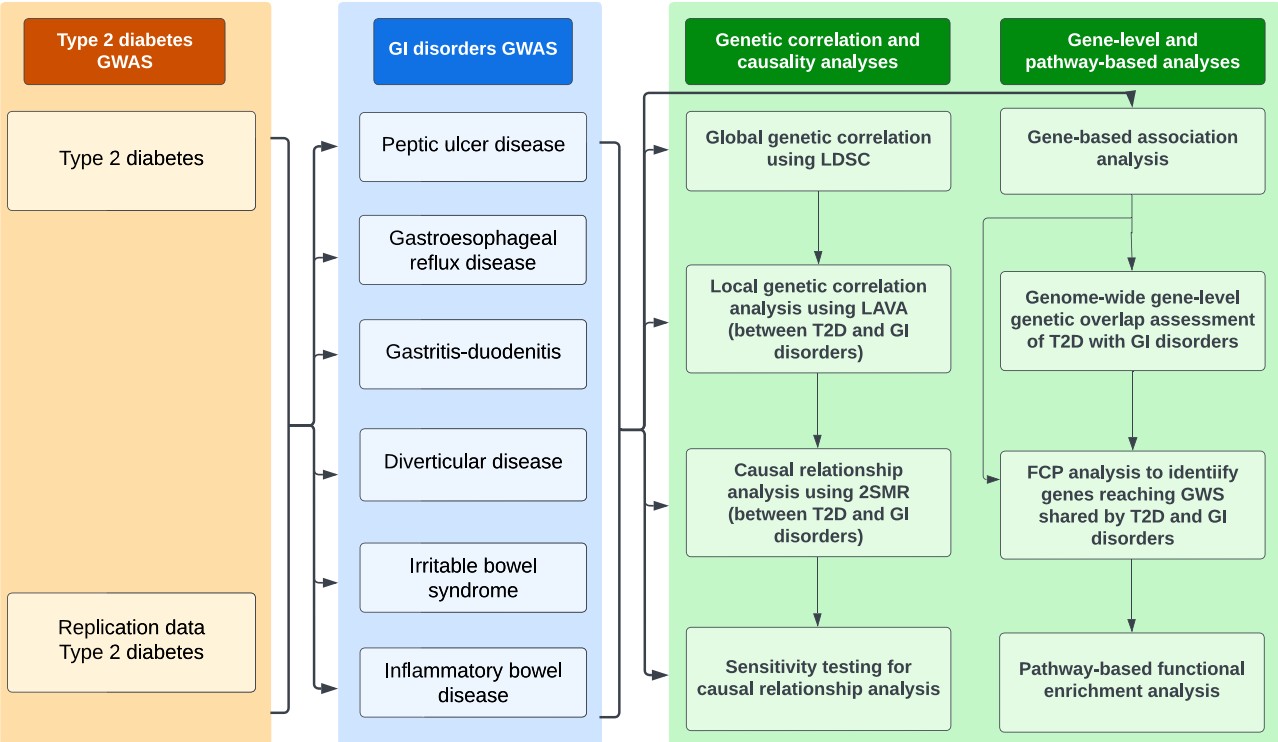

**Fig. 1 | Workflow summarising the study design and methods used in this study.** GI gastrointestinal, GWAS genome-wide association studies, FCP Fisher's combined *P* value, 2SMR two-sample Mendelian randomisation, LAVA local analysis of [co]variant association, LDSC linkage disequilibrium score regression, T2D type 2 diabetes. The figure was created using Lucidchart (https://lucid.app)[103].

## Genome-wide genetic correlation between T2D and GI disorders

Our cross-trait bivariate global genetic correlation analyses provide insights into the relationship between T2D and GI disorders. The results presented in Table 2 and Supplementary Data 2 demonstrate a consistently positive genetic correlation (rg) between T2D and all examined GI disorders, except IBD. Specifically, we observed a positive genetic correlation between T2D and PUD ($r_g = 0.29$, se = 0.04, $P = 2.03 \times 10^{-11}$), GERD ($r_g = 0.36$, se = 0.02, $P = 3.02 \times 10^{-59}$), gastritis-duodenitis ($r_g = 0.32$, se = 0.04, $P = 2.75 \times 10^{-17}$), IBS ($r_g = 0.13$, se = 0.04, $P = 1.24 \times 10^{-3}$), and diverticular disease ($r_g = 0.19$, se = 0.03, $P = 7.13 \times 10^{-12}$), but not with IBD ($r_g = 0.06$, se = 0.04, $P = 1.10 \times 10^{-1}$). Similarly, we found a highly significant global genetic correlation between T2D and PGM (Table 2). We replicated these significant positive genetic correlation findings using another set of GWAS summary data for GI disorders and T2D (Table 2 and Supplementary Data 2). Notably, excluding the major histocompatibility complex (MHC) region had little or no effect on our genetic correlation estimates or significance (Supplementary Data 2). Despite evidence of no considerable sample overlap across the GWAS for T2D and GI disorders (Supplementary Data 2), we did not constrain genetic covariance intercepts in the present study; hence, our genetic correlation estimates may be conservative.

Furthermore, we investigated the genetic correlation between T2D and GI disorders using the BMI-adjusted T2D GWAS summary data (T2D$_{bmiadj}$) to account for the potential effect of BMI on the relationship between these disorders. Our findings reveal a reduction in the strength of the genetic correlation between T2D and GI disorders following the use of data adjusted for BMI. However, T2D maintains a positive and significant genetic correlation with all GI disorders except IBD. For instance, we found a significant genetic correlation of T2D$_{bmiadj}$ with PUD ($r_g = 0.19$, se = 0.05, $P = 2.92 \times 10^{-5}$), GERD ($r_g = 0.24$, se = 0.03, $P = 5.11 \times 10^{-20}$), gastritis-duodenitis ($r_g = 0.25$, se = 0.04, $P = 5.23 \times 10^{-9}$), IBS ($r_g = 0.10$, se = 0.04, $P = 7.88 \times 10^{-3}$), PGM ($r_g = 0.25$, se = 0.04, $P = 4.29 \times 10^{-22}$) and diverticular disease ($r_g = 0.11$, se = 0.03, $P = 6.13 \times 10^{-4}$) [Supplementary Data 2]. The genetic correlation between T2D and IBD remained nonsignificant in the BMI-adjusted analysis ($r_g = 0.02$, se = 0.05, $P = 6.98 \times 10^{-1}$). We replicated these findings using additional GWAS data for GI disorders (Supplementary Data 2).

## Bivariate local genetic correlations between T2D and GI disorders

In addition to assessing genetic correlation across the genome (global correlation analysis), the present study aims to identify specific genomic regions contributing disproportionately to the relationship between T2D and GI disorders. Accordingly, we performed local genetic correlation analysis using LAVA software with a total of 13 pairwise bivariate tests conducted separately for T2D and PUD, 33 for T2D and GERD, 7 for T2D and gastritis-duodenitis, 19 for T2D and IBS, 13 for T2D and diverticular disease and 29 for T2D and IBD (Supplementary Data 2). Accounting for multiple testing, we considered $P < 1.52 \times 10^{-3}$ to be significant (Bonferroni adjustment for performing the maximum number of 33 bivariate tests [0.05/33]) and $P < 0.05$ to indicate nominal significance.

Our findings reveal 24 significant local genetic correlations between T2D and GI disorders at 12 loci (GRCh37/hg19) across chromosomes 3, 6, 7, 11, and 13 (Supplementary Data 3). With many local signals identified, five loci (961, 963, 964, 965, and 966) contributed disproportionately (as hotspots) to the genetic correlation between T2D and GI disorders. All these loci are on chromosome 6 (Fig. 2, Supplementary Data 3), including 961 (chr6: 31427210 – 32208901 [T2D – GERD, T2D – IBS, and T2D – diverticular disease], 964 (chr6: 32539568 – 32586784) [T2D – GERD, T2D – diverticular disease, and T2D – IBD], 963 (chr6: 32454578 – 32539567), 965 (chr6: 32586785 – 32629239), and 966 (chr6: 32629240 – 32682213) [T2D–GERD, T2D–gastritis-duodenitis, T2D–diverticular disease, and T2D–IBD] (Fig. 2 and Supplementary Data 3).

The local genetic correlations of T2D with GI disorders were positive across all the loci implicated, except for correlations of T2D with IBD, which were negative (Fig. 2). Additionally, most loci identified were replicated in the BMI-adjusted analysis (using the T2D$_{bmiadj}$ GWAS,

**Table 1 | GWAS summary data used for analysis**

| Data | Cases | Control | *N* | Ancestry | Phenotype source/definition |
|---|---|---|---|---|---|
| T2D (Mahajan et al.)[83] | 74,124 | 824,006 | 898130 | European | T2D data from the DIAGRAM consortium |
| T2D adjusted for BMI[83] | 74,124 | 824,006 | 898130 | | T2D data from the DIAGRAM consortium |
| GERD (UKB_QSKIN, An et al.)[84] | 71,522 | 261,079 | 332,601 | | Data from the UKB and the QSKIN study |
| PUD (Wu et al.)[43] | 16,666 | 439,661 | 456,327 | | UKB data code described in Wu et al. 2021 |
| Gastritis-duodenitis Phecode 535 (Lee Lab) | 28,941 | 378,124 | 407,065 | | Full White British samples from the Lee Lab |
| IBS (Wu et al.)[43] | 28,518 | 426,803 | 455,321 | | UKB data code described in Wu et al. 2021 |
| Diverticulosis and diverticulitis Phecode 562 (Lee Lab) | 27,311 | 334,783 | 362,094 | | Full White British samples from the Lee Lab |
| IBD (Wu et al.)[43] | 7045 | 449,282 | 456,327 | | UKB data code described in Wu et al. [43] |
| PGM (Wu et al.)[43] | 90,175 | 366,152 | 456,327 | | Phenotype combining PUD and GERD and/or corresponding medications/treatments. |
| *Data for further analyses, replication or adjustment in the multivariable MR analysis* | | | | | |
| GORD (Wu et al.)[43] | 54,854 | 401,473 | 456,327 | European | UKB data code described in Wu et al. [43] |
| PUD Phecode 531 (Lee Lab) | 7,436 | 401,525 | 408,961 | | Full White British samples from the Lee Lab |
| Gastritis-duodenitis (Watanabe et al.[85]) | 14,477 | 286,314 | 300,791 | | Main ICD10: K29 Gastritis and duodenitis |
| IBS Phecode 564.1 (Lee Lab) | 5,548 | 334,783 | 340,331 | | Full White British samples from the Lee Lab |
| Diverticular disease (Watanabe et al. [85]) | 14,028 | 286,763 | 300,791 | | Main ICD10: K57 Diverticular disease |
| IBD (Liu et al. 2015) | 12,882 | 21,770 | 34,652 | | Data from the IBD genetic consortium |
| Lansoprazole (Watanabe et al.[85]) | 13,559 | 266,884 | 280,443 | | UKB treatment/medication code lansoprazole |
| T2D (Xue et al.)[44] | 62,832 | 596,424 | 659256 | | A meta-analysis combining DIAGRAM, GERA, and the full cohort release of the UKB. |
| UKB T2D Phecode 250.2 (Lee Lab) | 18,945 | 388,756 | 407,701 | | Full British samples from the Lee Lab |
| BMI (GWAS Atlas ID: 3445)[85] | | | 379,831 | | GWAS data from the UKB |
| BMI (Yengo, et al.)[45] | | | 681,275 | | GWAS data from the GIANT Consortium |

*DIAGRAM* DIAbetes Genetics Replication and Meta-analysis, *GERA* Genetic Epidemiology Research on Aging, *UKB* United Kingdom Biobank, *ICD* International Classification of Diseases, *T2D* type 2 diabetes, *GERD (or GORD)* gastroesophageal reflux disease, *PGM* medicated phenotype for GERD and PUD, *PUD* peptic ulcer disease, *IBS* irritable bowel syndrome, *IBD* inflammatory bowel disease, *BMI* body mass index, *N* sample size. Phecode 562 comprises diverticulosis and diverticulitis, reflecting this study's definition of diverticular disease.

Supplementary Data 3), providing evidence that BMI does not account for the identified loci shared by T2D and GI disorders. We found that many of the identified loci (significant or nominally significant) were completely shared (that is, 'the 95% confidence interval for the explained variance included 1'[42]) by T2D and GI disorders (Table 3), further supporting evidence of their shared genetic basis. Three loci, 950 (chr6: 25684630 – 26396200), 1126 (chr7: 27351287–28890886), and 1719 (chr11: 112755447–113889019) demonstrated positive bivariate local genetic correlations between T2D and PUD (Table 3). While the loci were only nominally significant (Supplementary Data 3), all of them were entirely shared by the two traits (Table 3) and in the same direction as the estimate from the global genetic correlation (Fig. 2 and Table 3).

Similarly, we detected nine significant positive bivariate local genetic correlations between T2D and GERD at nine loci (464, 961, 963, 964, 965, 966, 969, 1667, 1895) and across chromosomes 3, 6, 11, and 13 (Supplementary Data 3). We also found seven nominally significant loci (954, 1126, 1719, 1957, 2209, 2281, and 2351) for T2D–GERD across chromosomes 6, 7, 11, 14, 17, 18, and 19 (Supplementary Data 3). Seven of these T2D–GERD-associated loci (significant or nominally significant) were entirely shared by the two traits (Table 3).

Our local genetic correlation findings are largely consistent with the LDSC-based global genetic correlation results, except for the association between T2D and IBD (Fig. 2). For instance, LDSC revealed positive and highly significant global genetic correlations between T2D and PUD, GERD, gastritis-duodenitis, diverticular disease, and IBS, as shown in Fig. 2 and Table 2. LAVA reproduced similar findings (nominally significant for T2D–PUD). Conversely, while LDSC found no evidence of a significant global genetic correlation between T2D and IBD, LAVA revealed strong and highly significant negative local genetic correlations between the two disorders (Fig. 2, Table 2, and Supplementary Data 2), indicating a complex

relationship between the disorders. These significant negative local correlations were primarily across loci on chromosome 6 and, nominally, on chromosomes 16, 17, and 19 (Fig. 2, Table 3, and Supplementary Data 3).

**Results of bi-directional Mendelian randomisation analysis**

We performed a bi-directional two-sample Mendelian randomisation (MR) analysis to assess the potential causal relationship between T2D and GI disorders. MR mimics randomised control trials for causality assessment and is underlain by three assumptions (Fig. 3).

We consider the results of our MR analysis significant at $P < 8.33 \times 10^{-3}$ (0.05/6, Bonferroni adjustment for testing six GI disorders) and nominally significant at $P < 0.05$. Figure 4 summarises the results of our MR analysis, where the primary model (IVW) was at least nominally significant. We present a detailed report on all the models and tests in Supplementary Data 4. Based on the IVW model, we found a significant causal effect of genetic predisposition to T2D on GERD and nominally on PUD and gastritis-duodenitis (Fig. 4). We found an OR of 1.03 (95% CI: 1.01–1.04, $P = 3.80 \times 10^{-3}$) for GERD, 1.04 (95% CI: 1.01–1.07, $P = 3.29 \times 10^{-2}$) for PUD, and 1.03 (95% CI: 1.01–1.04, $P = 3.28 \times 10^{-2}$) for gastritis-duodenitis per standard deviation in liability to T2D. The result for the causal effect of T2D on GERD was consistent in its positive direction across the weighted median, MR–Egger, and the MR-PRESSO (Mendelian Randomization Pleiotropy RESidual Sum and Outlier) models. However, only the MR-PRESSO result reached statistical significance (OR: 1.02 [95% CI: 1.01–1.04], $P = 4.05 \times 10^{-3}$). The MR–Egger intercept for this finding was not significantly different from zero ($P_{\text{pleiotropy-test}} = 0.29$), indicating no unbalanced pleiotropy (Supplementary Data 4). Similarly, there was no evidence for significant SNP-effect heterogeneity ($P_{\text{heterogeneity-test}} = 0.55$, Supplementary Data 4), ruling out a likely bias from pleiotropic SNPs. Consistent with this premise, the MR-PRSSO model also did not produce

**Table 2 | Global genetic correlation between type 2 diabetes and gastrointestinal disorders**

| T2D | GI disorders | Rg | Se | *P* |
|---|---|---|---|---|
| T2D (Mahajan 2018) | PUD (Wu et al.[43]) | 0.29 | 0.04 | 2.03E−11 |
| | GERD (UKB_QSKIN) | 0.36 | 0.02 | 3.02E−59 |
| | Gastritis-duodenitis (PheCode_535) | 0.32 | 0.04 | 2.75E−17 |
| | IBS-GWAS (Wu et al.[43]) | 0.13 | 0.04 | 1.24E−03 |
| | Diverticular disease (PheCode 562) | 0.19 | 0.03 | 7.13E−12 |
| | IBD (Wu et al.[43]) | 0.06 | 0.04 | 1.10E−01 |
| | PGM (Wu et al.[43]) | 0.37 | 0.02 | 3.12E−57 |
| *Replication for GI disorders* | | | | |
| | PUD (PheCode_531) | 0.42 | 0.07 | 3.34E−10 |
| | GORD (Wu et al.[43]) | 0.31 | 0.03 | 1.12E−29 |
| | Gastritis-duodenitis (Watanabe et al.) | 0.23 | 0.06 | 4.08E−05 |
| | IBS (PheCode 564.1) | 0.2 | 0.05 | 1.06E−04 |
| | Diverticular disease (K57-diagnosed) | 0.12 | 0.03 | 1.79E−04 |
| | IBD (Watanabe et al.) | −0.05 | 0.03 | 9.32E−02 |
| | Lansoprazole (Watanabe et al.) | 0.38 | 0.05 | 3.16E−14 |
| *Replication for T2D* | | | | |
| T2D (UKB) | PUD (Wu et al.[43]) | 0.44 | 0.06 | 2.03E−13 |
| | GERD (GERD: UKB_QSKIN) | 0.47 | 0.03 | 1.91E−54 |
| | Gastritis-duodenitis (PheCode 535) | 0.47 | 0.05 | 6.56E−23 |
| | IBS-GWAS (Wu et al.[43]) | 0.16 | 0.05 | 3.44E−03 |
| | Diverticular disease (PheCode_562) | 0.25 | 0.03 | 1.65E−14 |
| | IBD (Wu et al.[43]) | 0.07 | 0.05 | 1.87E−01 |
| | PGM (Wu et al.[43]) | 0.48 | 0.03 | 3.86E−54 |
| *Replication for GI disorders* | | | | |
| | PUD (PheCode_531) | 0.58 | 0.09 | 1.00E−10 |
| | GORD (Wu et al.[43]) | 0.4 | 0.04 | 2.02E−27 |
| | Gastritis-duodenitis (Watanabe et al.) | 0.35 | 0.07 | 2.88E−06 |
| | IBS (PheCode 564.1) | 0.28 | 0.07 | 6.47E−05 |
| | Diverticular disease (K57-diagnosed) | 0.15 | 0.04 | 1.91E−04 |
| | IBD (Watanabe et al.) | −0.02 | 0.04 | 6.89E−01 |
| | Lansoprazole (Watanabe et al.) | 0.52 | 0.07 | 5.62E−14 |

*T2D* type 2 diabetes, *GI* gastrointestinal, *GERD* gastroesophageal reflux disease, *PUD* peptic ulcer disease, *IBS* irritable bowel syndrome, *IBD* inflammatory bowel disease, *PGM* phenotype combining PUD and GERD and/or corresponding medications/treatments, *rg* global genetic correlation estimates, *se* standard error, *P p*-value, *UKB* United Kingdom Biobank. Note: replication set data may not be completely independent; hence, replication may be considered partial.

corrected causal estimates, as no outlier instruments were identified (Supplementary Data 4).

The weighted median and the MR-PRESSO models support the IVW-based T2D's causal effect on PUD (Fig. 4 and Supplementary Data 4). Cochran's Q-test and MR-Egger intercept indicate no evidence of heterogeneity ($P_{\text{heterogeneity-test}} = 0.91$) or unbalanced pleiotropy ($P_{\text{pleiotropy-test}} = 0.63$), respectively. MR-PRESSO did not identify outlier SNPs and, thus, produced no corrected causal estimates (Supplementary Data 4). Concerning the results for the causal effect of T2D on gastritis-duodenitis, we

found no evidence for significant heterogeneity ($P_{\text{heterogeneity-test}} = 0.82$), just as the MR-Egger intercept did not deviate significantly from zero, indicating no unbalanced pleiotropy ($P_{\text{pleiotropy-test}} = 0.10$). MR-PRESSO also did not identify outlier SNPs and produced no corrected causal estimates (Fig. 4 and Supplementary Data 4).

In a reverse analysis, we assessed GI disorders as exposure variables against T2D as the outcome variable. Genetic predisposition to GERD was significantly associated with T2D in the IVW model (OR: 1.19 [95% CI: 1.03–1.37], $P = 1.39 \times 10^{-2}$) [Fig. 4, and Supplementary Data 4] at the nominal level of significance. Sensitivity testing using the weighted median (OR: 1.15 [95% CI: 1.01–1.30], $P = 3.45 \times 10^{-2}$) and crude MR-PRESSO (OR: 1.19 [95% CI: 1.04–1.37], $P = 2.43 \times 10^{-2}$) methods support this result (Supplementary Data 4). Cochran's Q-test, however, identified a significant SNP effect heterogeneity ($P_{\text{heterogeneity-test}} = 5.33 \times 10^{-5}$, Supplementary Data 4), although the MR-Egger intercept was not significantly different from zero ($P_{\text{pleiotropy-test}} = 0.37$) [Supplementary Data 4]; hence, we prioritised the weighted median model's finding given that the method is more robust to heterogeneity. Consistent with the heterogeneity results, MR-PRESSO excluded outlier variants and estimated a corrected causal estimate that showed a marginally nonsignificant causal effect of GERD on T2D (OR: 1.11 [95% CI: 0.99–1.23], $P = 9.00 \times 10^{-2}$) [Fig. 4 and Supplementary Data 4]. Lastly, based on these MR models, we found no evidence for the causal relationship of T2D with diverticular disease, IBS, and IBD, irrespective of the direction of analysis (Supplementary Data 4).

### Results of multivariable MR analysis
Using multivariable MR analysis, we assessed the causal relationship of T2D with PUD, gastritis-duodenitis, and GERD (phenotypes with consistent significant results in the univariable MR analyses) adjusting for BMI. This analysis confirms the significant causal findings in the univariable MR, with adjustment for genetically predicted BMI, suggesting that although BMI is causally associated with T2D and the GI disorders included in this study (Supplementary Data 4), it does not explain the relationship between the disorders. For example, after adjusting for BMI, genetic liability to T2D was still causally associated with GERD (OR: 1.03, 95% CI: 1.01–1.06, $P = 1.58 \times 10^{-2}$), PUD (OR: 1.04, 95% CI: 1.01–1.08, $P = 4.00 \times 10^{-2}$), and gastritis-duodenitis (OR: 1.04, 95% CI: 1.01–1.07, $P = 1.50 \times 10^{-2}$), at the nominal level of significance (Supplementary Data 4). These findings were replicated using another set of T2D (ebi-a-GCST006867)[44] and BMI (ieu-b-40)[45] data with T2D remaining causally associated with GERD (OR: 1.04, 95% CI: 1.01–1.07, $P = 3.26 \times 10^{-3}$), and gastritis-duodenitis (OR: 1.06, 95% CI: 1.02–1.09, $P = 1.07 \times 10^{-3}$), and nominally with PUD (OR: 1.05, 95% CI: 1.01–1.09, $P = 2.04 \times 10^{-2}$) (Supplementary Data 4). Notably, the causal effect of T2D on gastritis-duodenitis, which was only nominally significant in the univariable analysis, attained our cut-off for significance status in the multivariable analysis. Similarly, after adjusting for BMI, in a bidirectional assessment, we found a nominally significant causal effect of GERD on T2D (OR: 1.24, 95% CI: 1.05–1.47, $P = 1.26 \times 10^{-2}$), replicated using a different T2D (ebi-a-GCST006867)[44] GWAS (OR: 1.31, 95% CI: 1.07–1.59, $P = 7.79 \times 10^{-3}$) and reaching a significant status (Supplementary Data 4).

### Gene-level analysis shows a significant overlap between T2D and GI disorders
Table 4 summarises our gene-level genetic overlap findings with the 'results explained' subsection (under the table), enabling additional insights into the relationships. Following gene mapping analysis in MAGMA, we identified equivalent protein-coding genes for pairs of T2D and GI disorders—18,710 (for T2D–PUD), 18,823 (for T2D–GERD), 19,118 (for T2D–gastritis-duodenitis), 19,118 (for T2D–diverticular disease), 18,710 (for T2D–IBS) and 18,710 (for T2D–IBD). At the threshold of $P_{\text{gene}} < 0.05$, a total of 517 genes overlapped between T2D and PUD, 1261 between T2D and GERD, 588 between T2D and gastritis-duodenitis, 975 between T2D and diverticular disease, 566 between T2D and IBS, and 711 between T2D and IBD (Table 4).

We estimated and compared the expected proportion of gene overlap with their corresponding observed proportion of gene overlap.

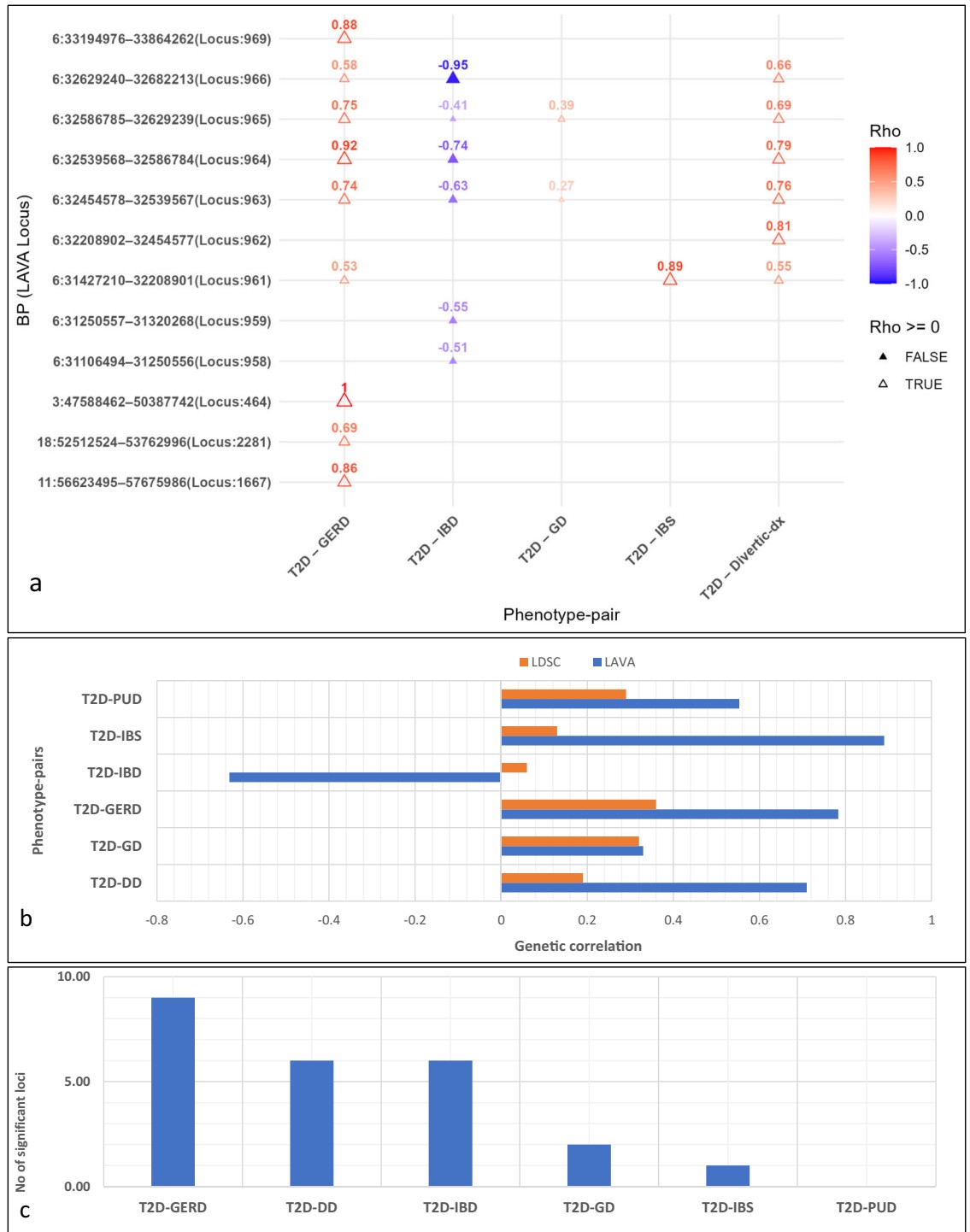

**Fig. 2 | Results of local genetic correlation analysis.** Chr chromosome, dx disease, IBS irritable bowel syndrome, DD diverticular disease, GERD gastroesophageal reflux disease, GD gastritis-duodenitis, IBD inflammatory bowel disease, *p* p-value, PUD peptic ulcer disease, T2D type 2 diabetes, BP base pair position (hg19), LAVA local analysis of [co]variant association, LDSC linkage disequilibrium score regression. Note: (i) the local genetic correlation between T2D and PUD did not survive multiple testing correlations but was nominally significant, (ii) the global genetic correlation between T2D and IBD was not significant, (iii) there was a significant local genetic correlation between T2D and IBD, but in the negative direction, unlike other

gastrointestinal disorders. **a** Loci contributing disproportionately to the genetic correlation of T2D with GI disorders. Note: Only results surviving multiple testing corrections are presented here (significance at $P < 1.52 \times 10^{-3}$, following Bonferroni correction for the maximum number of bivariate tests performed, [0.05/33], that is for T2D–GERD; this can be conservative for other T2D–GI disorders pairs, but we chose it for consistency). **b** Comparison of the global (LDSC) and mean local genetic correlations (LAVA) between T2D and gastrointestinal disorders. c: The number of significant local genetic correlations of T2D with GI disorders.

## Table 3 | Loci completely shared by T2D and GI disorders

| S/N | Locus | chr | Start | Stop | phen1 | phen2 | Rho | rho lower | rho upper | r2 | *P* |
|---|---|---|---|---|---|---|---|---|---|---|---|
| 1 | 464 | 3 | 47588462 | 50387742 | T2D | GERD | 1.00 | 0.50 | 1.00 | 1.00 | 1.24E−04 |
| 2 | 950 | 6 | 25684630 | 26396200 | T2D | PUD | 0.66 | 0.08 | 1.00 | 0.44 | 3.50E−02 |
| 3 | 954 | 6 | 29529756 | 29833843 | T2D | GERD | 0.64 | 0.13 | 1.00 | 0.41 | 2.25E−02 |
| 4 | 958 | 6 | 31106494 | 31250556 | T2D | Diverticular disease | 0.58 | 0.17 | 1.00 | 0.33 | 8.79E−03 |
| 5 | 960 | 6 | 31320269 | 31427209 | T2D | Diverticular disease | 0.67 | 0.25 | 1.00 | 0.44 | 3.53E−03 |
| 6 | 961 | 6 | 31427210 | 32208901 | T2D | IBS | 0.89 | 0.45 | 1.00 | 0.80 | 1.89E−04 |
| 7 | 962 | 6 | 32208902 | 32454577 | T2D | Diverticular disease | 0.81 | 0.54 | 1.00 | 0.65 | 5.07E−07 |
| 8 | 964 | 6 | 32539568 | 32586784 | T2D | GERD | 0.92 | 0.84 | 1.00 | 0.85 | 1.33E−30 |
| 9 | 966 | 6 | 32629240 | 32682213 | T2D | IBD | −0.95 | −1.00 | −0.81 | 0.91 | 2.70E−12 |
| 10 | 969 | 6 | 33194976 | 33864262 | T2D | GERD | 0.88 | 0.47 | 1.00 | 0.77 | 1.77E−04 |
| 11 | 1126 | 7 | 27351287 | 28890886 | T2D | PUD | 0.46 | 0.09 | 1.00 | 0.21 | 1.52E−02 |
| 12 | 1667 | 11 | 56623495 | 57675986 | T2D | GERD | 0.86 | 0.43 | 1.00 | 0.74 | 3.79E−04 |
| 13 | 1719 | 11 | 112755447 | 113889019 | T2D | PUD | 0.54 | 0.02 | 1.00 | 0.29 | 4.69E−02 |
|  | 1719 | 11 | 112755447 | 113889019 | T2D | GERD | 0.65 | 0.26 | 1.00 | 0.42 | 2.05E−03 |
| 14 | 1895 | 13 | 53336572 | 54684856 | T2D | GERD | 0.79 | 0.46 | 1.00 | 0.62 | 2.59E−05 |
| 16 | 1957 | 14 | 22760701 | 23985936 | T2D | IBS | 0.53 | 0.15 | 1.00 | 0.29 | 9.17E−03 |

Completely shared (95% confidence intervals [CIs] for the explained variance included 1[42]), *Chr* chromosome, *GERD* gastroesophageal reflux disease, *IBD* inflammatory bowel disease, *GD* gastritis-duodenitis, *IBS* irritable bowel syndrome, *p* p-value, *PUD* peptic ulcer disease, *T2D* type 2 diabetes, *phen* phenotype.

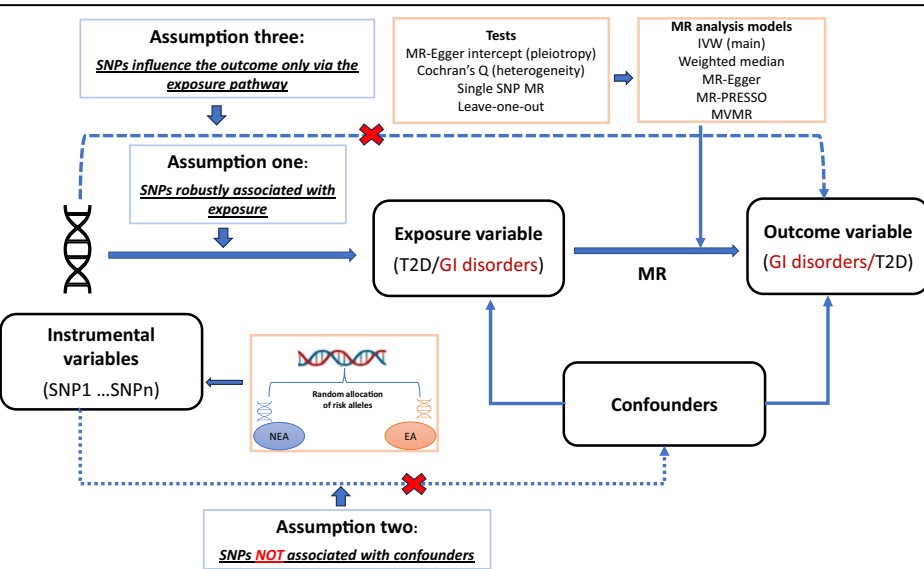

**Fig. 3 | Mendelian randomisation assumptions and schematic representation of study design.** The Figure illustrates Mendelian randomisation and its three underlying assumptions[30]: (1) Genetic variants (SNPs) used as instrumental variables are robustly associated with the exposure, (2) SNPs for the exposure variables are not associated with confounders, and (3) SNPs for the exposure variables influence the outcomes through no other pathway but the exposure. In the first round of analysis, we utilised T2D as exposure variables and GI disorders as outcomes. In the reverse analysis, we used GI disorders as exposures and T2D as the outcome variables. GI gastrointestinal, SNP single nucleotide polymorphism, T2D type 2 diabetes, MR Mendelian randomisation, EA effect allele, NEA non-effect allele, MVMR multivariable MR, MR-PRESSO Mendelian Randomization Pleiotropy RESidual Sum and Outlier.

Our results indicate that T2D demonstrated significant gene-level genetic overlap with all GI disorders, including PUD ($P_{\text{binomial-test}}$: $8.64 \times 10^{-6}$), GERD ($P_{\text{binomial-test}}$: $<2.20 \times 10^{-16}$), gastritis-duodenitis ($P_{\text{binomial-test}}$: $7.07 \times 10^{-8}$), diverticular disease ($P_{\text{binomial-test}}$: $<2.20 \times 10^{-16}$), IBS ($P_{\text{binomial-test}}$: $9.65 \times 10^{-11}$), and IBD ($P_{\text{binomial-test}}$: $3.51 \times 10^{-14}$) [Table 4]. To explore the potential contribution of BMI to this relationship, we further assessed gene-level genetic overlap using T2D data adjusted for BMI. Findings from this analysis reveal a slight reduction in the significance of our estimates; nonetheless, T2D maintains a statistically significant gene-level genetic overlap with all GI disorders (Table 4).

### Genome-wide significant (sentinel) genes shared by T2D and GI disorders

Our gene association analysis identified genome-wide significant (GWS) genes ($P_{\text{gene}} < 2.62 \times 10^{-6}$, that is, genes that were already GWS in our data, 'sentinel genes') with T2D having the highest number at 504

($P_{\text{gene-T2D}} < 2.62 \times 10^{-6}$, Supplementary Data 5), and 307 for T2D adjusted for BMI, revealing less number of SNPs when BMI was accounted for (Supplementary Data 5). For GI disorders, IBD has a total of 52 GWS genes ($P_{\text{gene-IBD}} < 2.62 \times 10^{-6}$, Supplementary Data 6), while diverticular disease has 50 ($P_{\text{gene-diverticular-disease}} < 2.62 \times 10^{-6}$, Supplementary Data 7). Others included 41 GWS genes for GERD ($P_{\text{gene-GERD}} < 2.62 \times 10^{-6}$, Supplementary Data 8), 11 for PUD ($P_{\text{gene-PUD}} < 2.62 \times 10^{-6}$, Supplementary Data 9), three (*RNF5, HLA-DQA1,* and *HLA-DQB1*) for gastritis-duodenitis ($P_{\text{gene-gastritis-duodenitis}} < 2.62 \times 10^{-6}$, Supplementary Data 10), and three (*HLA-C, PITPNM2,* and *MPHOSPH9*) for IBS ($P_{\text{gene-IBS}} < 2.62 \times 10^{-6}$, Supplementary Data 11).

Assessing overlap between GWS genes for T2D ($P_{\text{gene-T2D}} < 2.62 \times 10^{-6}$) and GWS genes for GI disorders ($P_{\text{gene-Gi-disorders}} < 2.62 \times 10^{-6}$), that is, sentinel genes, we found 11 (*TNXB, ATF6B, C6orf10, HLA-DRA, HLA-DQA1, HLA-DQB1, CARD9, C6orf47, SNAPC4, EHMT2,* and *HLA-DRB1*) were shared by both T2D and IBD (the highest) [Supplementary Data 12],

**Fig. 4 | Putative bidirectional causality between type 2 diabetes and gastrointestinal disorders.** MR Mendelian randomisation, T2D type 2 diabetes, PUD peptic ulcer disease, GERD gastroesophageal reflux disease, IVW inverse variance weighted, MR-PRESSO Mendelian randomisation pleiotropy residual sum and outlier, P P-value, OR odds ratio, CI confidence interval, nIV number of instrumental variables, [P] MR–Egger intercept p-value, P# The p-value for each of the model. Note: (1) We reported the corrected estimates in MR-PRESSO; however, we used the crude estimates where there are no corrected results. (2) In the present study, we used strict clumping parameters and manual removal of potentially pleiotropic SNPs (see Methods), which may result in a conservative but robust approach, and findings are unlikely to have been falsely positive.

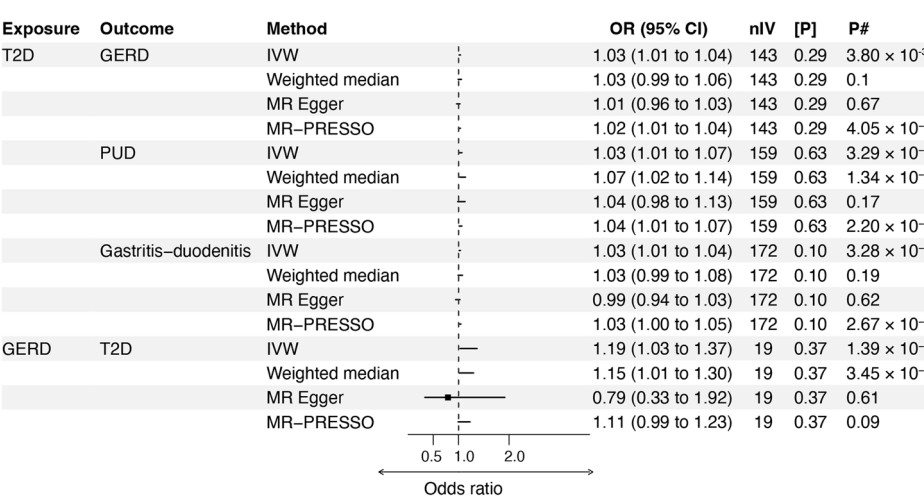

| Exposure | Outcome | Method | OR (95% CI) | nIV | [P] | P# |
|---|---|---|---|---|---|---|
| T2D | GERD | IVW | 1.03 (1.01 to 1.04) | 143 | 0.29 | $3.80 \times 10^{-3}$ |
| | | Weighted median | 1.03 (0.99 to 1.06) | 143 | 0.29 | 0.1 |
| | | MR Egger | 1.01 (0.96 to 1.03) | 143 | 0.29 | 0.67 |
| | | MR–PRESSO | 1.02 (1.01 to 1.04) | 143 | 0.29 | $4.05 \times 10^{-3}$ |
| | PUD | IVW | 1.03 (1.01 to 1.07) | 159 | 0.63 | $3.29 \times 10^{-2}$ |
| | | Weighted median | 1.07 (1.02 to 1.14) | 159 | 0.63 | $1.34 \times 10^{-2}$ |
| | | MR Egger | 1.04 (0.98 to 1.13) | 159 | 0.63 | 0.17 |
| | | MR–PRESSO | 1.04 (1.01 to 1.07) | 159 | 0.63 | $2.20 \times 10^{-2}$ |
| | Gastritis–duodenitis | IVW | 1.03 (1.01 to 1.04) | 172 | 0.10 | $3.28 \times 10^{-2}$ |
| | | Weighted median | 1.03 (0.99 to 1.08) | 172 | 0.10 | 0.19 |
| | | MR Egger | 0.99 (0.94 to 1.03) | 172 | 0.10 | 0.62 |
| | | MR–PRESSO | 1.03 (1.00 to 1.05) | 172 | 0.10 | $2.67 \times 10^{-2}$ |
| GERD | T2D | IVW | 1.19 (1.03 to 1.37) | 19 | 0.37 | $1.39 \times 10^{-2}$ |
| | | Weighted median | 1.15 (1.01 to 1.30) | 19 | 0.37 | $3.45 \times 10^{-2}$ |
| | | MR Egger | 0.79 (0.33 to 1.92) | 19 | 0.37 | 0.61 |
| | | MR–PRESSO | 1.11 (0.99 to 1.23) | 19 | 0.37 | 0.09 |

Odds ratio (0.5  1.0  2.0)

## Table 4 | Gene-level genetic overlap of T2D with GI disorders

| Discovery set | | | Target set | | | Number of genes overlapping the discovery and the target sets at $P_{gene} < 0.05$ | Proportion of gene overlap | | Binomial test |
|---|---|---|---|---|---|---|---|---|---|
| T2D | Total number of genes in the discovery set | Number of genes in the discovery set at $P_{gene} < 0.05$ | GI disorders | Total number of genes in the target set (GI disorders) | Number of genes in the target set at $P_{gene} < 0.05$ | | Expected | Observed (%) | P Value |
| *Gene-level genetic overlap between T2D and GI disorders* | | | | | | | | | |
| T2D(Mahajan 2018) | 18,710 | 5359 | PUD | 18,710 | 1541 | 517 | 0.285 | 0.336 (33.6) | $8.64 \times 10^{-6*}$ |
| | 18,823 | 5378 | GERD | 18,823 | 3343 | 1261 | 0.286 | 0.377 (37.7) | $<2.2 \times 10^{-16}$ |
| | 19,118 | 5403 | GastritisD | 19,118 | 1722 | 588 | 0.283 | 0.341 (34.1) | $7.07 \times 10^{-8}$ |
| | 19,118 | 5402 | Diverticular dx | 19,118 | 2529 | 975 | 0.283 | 0.386 (38.6) | $<2.20 \times 10^{-16}$ |
| | 18,710 | 5359 | IBS | 18,710 | 1575 | 566 | 0.285 | 0.359 (35.9) | $9.65 \times 10^{-11}$ |
| | 18,710 | 5359 | IBD | 18,710 | 1956 | 711 | 0.285 | 0.363 (36.3) | $3.51 \times 10^{-14}$ |
| *Gene-level genetic overlap between BMI-adjusted T2D and GI disorders* | | | | | | | | | |
| T2D (BMI-adjusted) | 18,710 | 4077 | PUD | 18,710 | 1541 | 367 | 0.218 | 0.238 (23.8) | 0.03 |
| | 18,823 | 4137 | GERD | 18,823 | 3343 | 936 | 0.220 | 0.280 (28.0) | $2.41 \times 10^{-16}$ |
| | 19,107 | 4256 | Gastritis-D | 19,107 | 1723 | 459 | 0.223 | 0.266 (26.6) | $1.26 \times 10^{-5}$ |
| | 19105 | 4255 | Diverticular dx | 19,105 | 2512 | 751 | 0.223 | 0.299 (29.9) | $2.20 \times 10^{-16}$ |
| | 18710 | 4077 | IBS | 18,710 | 1575 | 430 | 0.218 | 0.273 (27.3) | $1.51 \times 10^{-7}$ |
| | 18710 | 4077 | IBD | 18,710 | 1956 | 555 | 0.218 | 0.284 (28.4) | $5.28 \times 10^{-12}$ |

*T2D* type two diabetes, *GI* gastrointestinal tract disorders, *PUD* peptic ulcer disease, *GERD* gastroesophageal reflux disease, *gastris-D* gastritis-duodenitis, *IBS* irritable bowel syndrome, *IBD* inflammatory bowel disease, *T2D_{bmiadj}* T2D adjusted for body mass index.

* Results explained: In the gene-level overlap analysis, we aimed to assess whether the proportion of overlapping genes was more than expected by chance. Thus, we estimated and compared the observed proportion of gene overlap with the expected proportions (the null). Here, we demonstrate one of the results where we examined the overlap between T2D and PUD*. In this example, we defined the 'observed proportion of gene overlap' ('$O_{obs}$') as equal to 'the number of genes overlapping' T2D and PUD at $P_{gene} < 0.05$ (that is, 517) divided by 'the total number of genes associated with the target set' (PUD) at $P_{gene} < 0.05$ (that is, 1541). Expressed mathematically, $O_{obs} = 517/1541$. On the other hand, the 'expected proportion of gene overlap' (*e*) is equal to 'the number of genes associated with the discovery set' (T2D) at $P_{gene} < 0.05$ (that is, 5359) divided by the 'total number of genes' (that is, 18,710) [mathematically, $e = 5359/18710$]. We tested whether the observed proportion of genes was more than expected by chance by applying a one-sided exact binomial test in the R statistical platform [binom.test (517, 1541,0.285, alternative = c("greater")). The P value of $8.64 \times 10^{-6}$ indicates that the proportion of overlap between T2D and PUD was more than expected by chance.

six (*AGER, EHMT2, HLA-DRB1, HLA-DRB5, FAM185A, and TRPS1*) by T2D and diverticular disease (Supplementary Data 13), three (*RNF5, HLA-DQA1, and HLA-DQB1*) by T2D and gastritis-duodenitis (Supplementary Data 14), three (*RBM6, RBM5, and TCF4*) by T2D and GERD (Supplementary Data 14), two (*MPHOSPH9, and PITPNM2*) by T2D and IBS (Supplementary Data 14), and one (*SLC22A3*) by T2D and PUD [Supplementary Data 14]. Two GWS genes (*EHMT2 and HLA-DRB1*) overlapped between T2D, IBD, and diverticular disease (Supplementary Data 15). Additionally, we found two overlapping genes (*HLA-DQA1 and HLA-DQB1*) across T2D, IBD, and gastritis-duodenitis (Supplementary Data 15).

Using the BMI-adjusted T2D, the number of overlapping GWS genes decreased slightly; for example, we found nine (*TNXB, ATF6B, C6orf10, HLA-DRA, HLA-DQA1, HLA-DQB1, CARD9, EHMT2, and HLA-DRB1*) sentinel genes ($P_{gene-T2D-bmiadj} < 2.62 \times 10^{-6}$ and $P_{gene-Gi-disorders} < 2.62 \times 10^{-6}$) shared by both T2D_{bmiadj} and IBD (Supplementary Data 12), three (*AGER, EHMT2, and HLA-DRB1*) by T2D_{bmiadj} and diverticular disease (Supplementary Data 13), two (*MPHOSPH9, and PITPNM2*) by T2D_{bmiadj} and IBS (Supplementary Data 14), one (*TCF4*) by T2D_{bmiadj} and GERD (Supplementary Data 14), and one (*SLC22A3*) by T2D_{bmiadj} and PUD [Supplementary Data 14].

## Shared genes reaching genome-wide significance for T2D and GI disorders

Given their significant SNP-level (genetic correlation) as well as gene-based genetic overlap, we performed a further assessment using the Fisher's Combined $P$ value (FCP) method (as in previous studies[31,32,46]) to identify genes shared by T2D and GI disorders across three broad categories: (1) GWS T2D ($P_{\text{gene-T2D}} < 2.62 \times 10^{-6}$) genes that were also associated with GI disorders (at $P_{\text{gene-GI-disorder}} < 0.05$), (2) GWS GI disorders ($P_{\text{gene-GI-disorder}} < 2.62 \times 10^{-6}$) genes that were similarly associated with T2D (at $P_{\text{gene-T2D}} < 0.05$), and (3) nominally significant genes that were not GWS in T2D ($0.05 < P_{\text{gene-T2D}} > 2.62 \times 10^{-6}$) or in GI disorders ($0.05 < P_{\text{gene-GI-disorder}} > 2.62 \times 10^{-6}$) but reached GWS status after the FCP analysis ($P_{\text{FCP}} < 2.62 \times 10^{-6}$)—that is, putatively novel genes shared by T2D and GI disorders (based on our data).

In the first category, we found 44 GWS genes for T2D ($P_{\text{gene-T2D}} < 2.62 \times 10^{-6}$) that were also associated with PUD ($P_{\text{gene-PUD}} < 0.05$) [Supplementary Data 16], 180 associated with GERD ($P_{\text{gene-GERD}} < 0.05$) [Supplementary Data 17], 89 with gastritis-duodenitis ($P_{\text{gene-gastritis-duodenitis}} < 0.05$) [Supplementary Data 18], 129 with diverticular disease ($P_{\text{gene-diverticular-disease}} < 0.05$) [Supplementary Data 19], 87 with IBS ($P_{\text{gene-IBS}} < 0.05$) [Supplementary Data 20], and 93 with IBD ($P_{\text{gene-IBD}} < 0.05$) [Supplementary Data 21]. Using the T2D adjusted for BMI, the number of overlapping genes was slightly lower. For example, we found 30 GWS genes for T2D$_{\text{bmiadj}}$ ($P_{\text{gene-T2D-bmiadj}} < 2.62 \times 10^{-6}$) that were also associated with PUD ($P_{\text{gene-PUD}} < 0.05$) [Supplementary Data 16], 105 associated with GERD ($P_{\text{gene-GERD}} < 0.05$) [Supplementary Data 17], 63 with gastritis-duodenitis ($P_{\text{gene-gastritis-duodenitis}} < 0.05$) [Supplementary Data 18], 90 with diverticular disease ($P_{\text{gene-diverticular-disease}} < 0.05$) [Supplementary Data 19], 56 with IBS ($P_{\text{gene-IBS}} < 0.05$) [Supplementary Data 20] and 65 with IBD ($P_{\text{gene-IBD}} < 2.62 \times 10^{-6}$) [Supplementary Data 21].

In the second category, one GWS ($P_{\text{gene-PUD}} < 2.62 \times 10^{-6}$, sentinel) gene for PUD (*SLC22A3*), 17 for GERD ($P_{\text{gene-GERD}} < 2.62 \times 10^{-6}$, *RBM6, RBM5, SEMA3F, MAML3, HIST1H3C, HIST1H1T, HLA-C, HLA-B, SETBP1, PDE4B, RABGAP1L, SGCD, ZNF322, FOXP2, DCC, TCF4,* and *CRTC1*) and three for gastritis-duodenitis ($P_{\text{gene-gastritis-duodenitis}} < 2.62 \times 10^{-6}$, *HLA-DQA1, HLA-DQB1, RNF5*) were associated with T2D ($P_{\text{gene-T2D}} < 0.05$) [Supplementary Data 22 - 24]. Similarly, we found 24 GWS ($P_{\text{gene-diverticular-disease}} < 2.62 \times 10^{-6}$) genes for diverticular disease, two for IBS ($P_{\text{gene-IBS}} < 2.62 \times 10^{-6}$, *HLA-C and PITPNM2*), and 26 for IBD ($P_{\text{gene-IBD}} < 2.62 \times 10^{-6}$) that were associated with T2D ($P_{\text{gene-T2D}} < 0.05$) [Supplementary Data 25–27]. Using the T2D adjusted for BMI, we observed a slight reduction in the number of overlapping genes or their significance, as documented in Supplementary Data 22–27.

In the third category, we identified many genes reaching GWS in the FCP analysis ($P_{\text{FCP}} < 2.62 \times 10^{-6}$), which were not GWS in either T2D ($0.05 < P_{\text{gene-T2D}} > 2.62 \times 10^{-6}$) or GI disorders GWAS ($0.05 < P_{\text{gene-GI-disorder}} > 2.62 \times 10^{-6}$), suggesting that they are putatively novel (based on our data). These included 15 genes for T2D–PUD ($P_{\text{FCP}} < 2.62 \times 10^{-6}$, Supplementary Data 28), 96 genes for T2D–GERD ($P_{\text{FCP}} < 2.62 \times 10^{-6}$, Supplementary Data 29), 20 for T2D–gastritis-duodenitis ($P_{\text{FCP}} < 2.62 \times 10^{-6}$, Supplementary Data 30), 78 for T2D–diverticular disease ($P_{\text{FCP}} < 2.62 \times 10^{-6}$, Supplementary Data 31), 28 for T2D–IBS ($P_{\text{FCP}} < 2.62 \times 10^{-6}$, Supplementary Data 32), and 48 for T2D–IBD ($P_{\text{FCP}} < 2.62 \times 10^{-6}$, Supplementary Data 33). In the analysis with T2D$_{\text{bmiadj}}$ data, we observed a slight reduction in the number and significance of overlapping genes, as shown in Supplementary Data 28–33.

Furthermore, we explored genes reaching GWS ($P_{\text{FCP}} < 2.62 \times 10^{-6}$) in their respective FCP analysis of T2D across two or more GI disorders. For example, we identified 9 GWS genes (FCP analysis) shared across T2D, PUD, and gastritis-duodenitis (Supplementary Data 34); 15 across T2D, PUD, and GERD (Supplementary Data 35); 12 shared by T2D, PUD, and diverticular disease (Supplementary Data 36); and six shared by T2D, GERD, PUD, and diverticular disease (Supplementary Data 37). Further, we identified three genes shared across T2D and all GI disorders (*LST1, ATP6V1G2-DDX39B, HLA-DQA1*) reaching GWS in their respective FCP

analysis, and 17 shared by T2D and all GI disorders except PUD (Supplementary Data 38). We also identified 19 genes shared across T2D, GERD, diverticular disease, IBS, and IBD (Supplementary Data 39).

## Biological pathways significantly enriched for T2D and GI disorders

We performed pathway-based functional enrichment analyses using genes overlapping T2D$_{\text{bmiadj}}$ and GI disorders (Supplementary Data 40–51) to identify biological pathways, mechanisms, or processes shared by T2D and GI disorders. Notably, several pathways or processes were significantly enriched, suggesting that they play roles in the underlying mechanisms of the disorders and, potentially, their comorbid state (Supplementary Data 52–57). Across several of the pathway-based analysis results, we found a significant enrichment of proinflammatory and immune-related mechanisms such as 'T-helper (Th)1 and Th2 cell differentiation' for T2D across GERD (adjusted $P$ value [$P_{\text{Adj}}$] = $1.44 \times 10^{-3}$), IBS ($P_{\text{Adj}} = 4.23 \times 10^{-2}$), diverticular disease ($P_{\text{Adj}} = 3.36 \times 10^{-3}$), and IBD ($P_{\text{Adj}} = 5.01 \times 10^{-7}$) [Supplementary Data 52–53, and 56–57]. Similarly, 'human T-cell leukemia virus 1 infection' was significantly enriched for genes shared by T2D and GERD ($P_{\text{Adj}} = 9.48 \times 10^{-5}$), T2D–IBS ($P_{\text{Ad}} = 8.53 \times 10^{-4}$), T2D–diverticular disease ($P_{\text{Adj}} = 3.34 \times 10^{-2}$), and T2D–IBD ($P_{\text{Adj}} = 9.84 \times 10^{-6}$) [Supplementary Data 52–53, and 56–47].

Reinforcing the putative involvement of the immune-mediated mechanisms in T2D, GI disorders, and potentially their comorbidity, our analyses implicate traits known for autoimmune mechanisms, including type 1 diabetes (T2D–GERD, T2D–diverticular disease, T2D–IBD, and T2D–IBS), asthma (T2D–IBD and T2D–IBS), inflammatory bowel disease (T2D–GERD, T2D–IBD, and T2D–IBS) and 'autoimmune thyroid disease' (T2D–IBS) [Supplementary Data 52–53, and 56–57]. Moreover, several biological pathways overrepresented for T2D relate to the MHC region across five of the six GI disorders (excluding PUD). For example, the 'MHC class II protein complex' was significantly enriched for genes overlapping T2D and GERD ($P_{\text{Adj}} = 5.87 \times 10^{-4}$), gastritis-duodenitis ($P_{\text{Adj}} = 3.00 \times 10^{-3}$), IBS ($P_{\text{Adj}} = 7.77 \times 10^{-5}$), diverticular disease ($P_{\text{Adj}} = 1.98 \times 10^{-3}$), and IBD ($P_{\text{Adj}} = 2.56 \times 10^{-7}$) [Supplementary Data 52–54, and 56–57].

Other MHC-related pathways, including 'MHC class II receptor activity' (T2D–diverticular disease, T2D–IBD, and T2D–IBS), 'MHC protein complex' and 'MHC protein complex assembly' (T2D–GERD, T2D–diverticular disease, T2D–IBD, and T2D–IBS), were also significantly enriched. Pathways related to antigen processing and presentation were similarly overrepresented for T2D across many GI disorders (GERD, IBS, diverticular disease, and IBD). Additionally, we found mechanisms of viral infections, including viral myocarditis, overrepresented for T2D–GERD ($P_{\text{Adj}} = 8.31 \times 10^{-3}$), T2D–IBS ($P_{\text{Adj}} = 1.50 \times 10^{-3}$), T2D–diverticular disease ($P_{\text{Adj}} = 8.25 \times 10^{-3}$), and T2D–IBD ($P_{\text{Adj}} = 4.88 \times 10^{-8}$), 'Ebola virus infection in host' for T2D–IBS ($P_{\text{Adj}} = 4.15 \times 10^{-3}$) and T2D–IBD ($P_{\text{Adj}} = 4.64 \times 10^{-5}$), and human papillomavirus infection for T2D–IBS ($P_{\text{Adj}} = 3.74 \times 10^{-2}$). Thyroid hormone signalling ($P_{\text{Adj}} = 3.18 \times 10^{-3}$), interferon signalling ($P_{\text{Adj}} = 8.68 \times 10^{-3}$), and notch signalling ($P_{\text{Adj}} = 1.57 \times 10^{-2}$) were uniquely associated with genes overlapping T2D and IBS (and partly, T2D and IBD), just as the leptin signalling pathway ($P_{\text{Adj}} = 4.43 \times 10^{-4}$) was implicated for T2D–IBD overlapping genes (Supplementary Data 52–53, and 56–57).

Biological pathways significantly enriched for genes overlapping between T2D and PUD (Supplementary Data 55) were different (to a great extent) from those of other GI traits. For example, mechanisms related to abnormal blood or urine chemistry, including abnormal circulating calcium ($P_{\text{Adj}} = 2.80 \times 10^{-4}$) and hypercalcemia ($P_{\text{Adj}} = 6.22 \times 10^{-4}$), were among the most significantly enriched for T2D and PUD (Supplementary Data 55). None of these pathways was significantly enriched in other GI disorders, except in T2D-gastritis-duodenitis (hypercalcemia [$P_{\text{Adj}} = 4.55 \times 10^{-2}$] and abnormal circulating calcium concentration [$P_{\text{Adj}} = 3.09 \times 10^{-2}$]) [Supplementary Data 42], possibly, indicating the biological similarity of PUD with gastritis-duodenitis. Other mechanisms overrepresented for T2D–PUD

include abnormal blood concentration of inorganic cations ($P_{Adj} = 1.92 \times 10^{-3}$), abnormality of urine calcium concentration ($P_{Adj} = 1.25 \times 10^{-2}$), hypercalciuria ($P_{Adj} = 5.40 \times 10^{-3}$) and proteinuria ($P_{Adj} = 4.18 \times 10^{-2}$) [Supplementary Data 55]. We equally found cardiovascular-related mechanisms, including 'aortic arch aneurysm', 'renovascular hypertension', and 'supravalvular aortic stenosis' (in T2D–PUD, Supplementary Data 55), and 'abnormal QT interval' as well as 'regulation of cardiac muscle cell action potential' overrepresented for T2D–gastritis-duodenitis (Supplementary Data 54).

Given several pathways linked to the MHC region were among those overrepresented in this study, largely implicating pro-inflammatory and immunological-related mechanisms, we assessed whether excluding the region made any difference to our results or conclusion. Our findings suggest that excluding the MHC region does not necessarily rule out the involvement of the immune or inflammatory pathways among the biological mechanisms shared by T2D and GI disorders. For example, while all MHC pathways were no longer significantly enriched in the analysis excluding the region, inflammatory or immunological mechanisms or processes were still overrepresented. These pathways include 'cytokine receptor binding', 'neurotrophin receptor binding', 'cellular response to peptide hormone stimulus', 'regulation of leukocyte differentiation', 'T cell differentiation', 'Th1 and Th2 cell differentiation', 'interleukin-11 signalling pathway', 'DNA damage response (only ATM dependent)', 'notch signalling pathway', and 'mechanoregulation and pathology of YAP/TAZ via hippo and non-hippo mechanisms', for genes overlapping T2D and IBD (Supplementary Data 57). For genes overlapping T2D and diverticular disease, two pathways known for potential inflammatory or immune-related functions were significantly enriched, namely, 'transforming growth factor beta (TGF-beta) binding' and 'TGF-beta signalling pathway' (Supplementary Data 56). Further, 'thyroid hormone signalling pathway', 'Notch signalling pathway', and 'antiviral mechanism by IFN-stimulated genes' were significantly enriched for genes overlapping T2D and IBS (Supplementary Data 53), while 'primary hypercortisolism' (having a potential implication in immune or inflammatory processes) were enriched for genes overlapping T2D and gastritis-duodenitis (Supplementary Data 54).

Lastly, with the exclusion of the MHC region, pathways related to the cardiovascular system were significantly enriched, particularly for genes overlapping T2D and IBS (atrial septum development and atrial septum morphogenesis, cardiac septum development and cardiac septum morphogenesis, cardiac atrium development and cardiac atrium morphogenesis, cardiac chamber development and cardiac chamber morphogenesis, heart morphogenesis, abnormal cardiac ventricular function, and transient ischemic attack). Others include dilated cardiomyopathy (T2D and GERD), regulation of cardiac muscle cell action potential, abnormal QT interval, and abnormal electrocardiogram (T2D and gastritis-duodenitis) [Supplementary Data 52–57]. We also found bone development-related pathways, including skeletal system morphogenesis and osteoblast differentiation (T2D and diverticular disease) and bone development (T2D and GERD) [Supplementary Data 52–57].

## Discussion

Several reports have linked diabetes with GI traits[5,8–10,18–25]. However, the potential shared genetic architecture and causal association of T2D with GI disorders and their underlying biological mechanisms remain unresolved. Here, we analyse large-scale GWAS summary data and provide comprehensive evidence for the genetic relationship between T2D and six common GI disorders. Our study reveals a highly significant genome-wide genetic correlation between T2D and GERD, PUD, gastritis-duodenitis, diverticular disease, and IBS, with and without excluding the MHC region. Gene-level genetic overlap results were consistent with these findings, further supporting evidence of a genetic relationship between T2D and GI disorders. Conventional observational studies have previously suggested a heightened risk for GERD[18], IBD[23,24], PUD[21,22], and IBS[25] in individuals with diabetes. Our finding of significant genetic correlation and gene overlap provides new

insights (to the best of our knowledge) into their relationships, implicating shared genetic susceptibility of T2D with these GI disorders.

Unlike other GI disorders, we found no evidence of a significant genome-wide genetic correlation between T2D and IBD. This result contrasts with previous observational studies that have suggested a risk-increasing relationship between IBD and T2D[5,23,24]. Our finding does not necessarily rule out a connection between IBD and T2D; however, it may indicate a complex genetic relationship between the two disorders, as previously reported in IBD's association with Alzheimer's disease[31,32,47] and cognitive traits[31]. Genetic correlations can reflect several causal structures, including instances where significant estimates are due to causal or biological pleiotropy[48]. Moreover, two traits can exhibit discordant correlations at different genomic locations, with the potential for negligible (or non-significant) results in the global correlation estimate, which averages effects across the genome[48]. The latter scenario may be the case in the present study, and some risk genetic variants for T2D likely protect against IBD and vice versa.

We conducted local genetic correlation analyses to gain further insights into the connection between T2D and GI disorders[42]. The local approach tests for signals at specific locations across the genome, making it more informative than the genome-wide correlation estimates. Our analyses yielded noteworthy findings, including substantial evidence of local genetic correlations between T2D and IBD (which was not significant in the global correlation results), primarily across several loci in chromosome 6 and, nominally significant, in chromosomes 16, 17, and 19. The identified regions contribute disproportionately to the relationship between the two conditions, and relying solely on the global correlation analysis would have obscured these notable signals. Our results thus underscore the importance of complementing findings from the global genetic correlation analysis with those from the local approach[42,48]. The local genetic correlation analysis results are consistent with the significant gene-level genetic overlap between T2D and IBD. Although the gene-overlap method does not distinguish the direction of effects, its results indicate that T2D and IBD share more genes than expected by chance, supporting the finding (from the local correlation approach) that certain genomic regions drive the association between the two disorders.

However, the local genetic correlations between T2D and IBD were negative, contrasting with the other GI disorders but substantiating that IBD demonstrates a more complex aetiological relationship with T2D (that is, not straightforward or linear, unlike global correlation estimates). Current findings may also suggest that IBD is aetiologically different from the other digestive traits assessed in this study, an observation previously noted[31,32,43] and supported in our research (Supplementary Data 2). For example, PUD, GERD, IBS, gastritis-duodenitis, and diverticular disease demonstrate moderate to strong positive global genetic correlations with each other (Supplementary Data 2), revealing some levels of strong genetic similarity between them. However, IBD behaves differently with little or no evidence of positive genetic association with the other GI traits[43] (Supplementary Data 2). Furthermore, we detected only three nominally significant positive local genetic correlations between T2D and PUD (at chromosomes 6, 7, and 11), unlike the significant global genetic correlation estimated using the LDSC method. All three local loci identified were completely shared by T2D and PUD, providing additional evidence of genetic overlap between the two disorders. The findings may also suggest that genetic effects between T2D and PUD are not localised but widespread across the genome, potentially differentiating PUD or its relationship with T2D from the rest of the GI disorders.

The consistent results of genetic correlations (global and local) and gene-level overlap between T2D and GI disorders suggest shared genetic susceptibility but could also indicate a putative causal relationship between the disorders. Accordingly, we used bidirectional MR analyses to investigate the potential causal link between these disorders. Overall, findings from the univariable MR analyses suggest the causal influence of T2D on PUD and gastritis-duodenitis and a bidirectional causal association with GERD. These results align with previous MR studies that have reported a significant

causal relationship between T2D and GERD, gastric ulcer, and gastritis[33,35]. However, we found no evidence for a significant causal association between T2D and IBS, diverticular disease, and IBD, unlike previous studies[34,35]. This discrepancy may be due to differences in the data analysed or measures applied across studies. Our analysis, for example, used stringent criteria, including strict LD clumping parameters and manual removal of suspected pleiotropic SNPs, to avoid false positives, which may contribute to conservative but robust findings.

We assessed the potential influence of BMI on the genetic relationship between T2D and GI disorders using analysis based on BMI-adjusted T2D data and statistical adjustment using the MR analysis method. In the BMI-adjusted T2D data analysis, the strength of our SNP-based genetic correlations and the gene-level overlap estimates decreased but remained statistically significant. The findings suggest that while there is a genetic association between T2D and GI disorders that partially reflects an underlying association with BMI, there is also a genetic overlap between the disorders not accounted for by BMI. Following adjustment for BMI in multivariable MR analyses, T2D remained causally associated with GERD, PUD, and gastritis-duodenitis. While BMI is also associated with GI disorders in our study, the present results agree with findings for our genetic overlap assessment and reports in a recent MR study[35] supporting a genetic association between T2D and GI disorders independent of BMI.

Next, we focused on identifying shared genes and biological pathways to enhance our understanding of the underlying mechanisms of these disorders. We found several genes shared across T2D and GI disorders (especially IBD, diverticular disease, and IBS), many of which were represented in the MHC region and known for immune-mediated functions[49]. Consistent with this result, our local genetic correlation analysis uncovered five loci (961, 963, 964, 965, 966) in the MHC region, contributing disproportionately to the genetic relationship of T2D with GERD, IBD, gastritis-duodenitis, diverticular disease, and IBS. These findings support (auto) immune mechanisms in the biology of T2D and the GI disorders. We found other shared genes outside the MHC region previously implicated in diabetes or GI disorders. For instance, variants of *TCF4* (shared across T2D, GERD, gastritis-duodenitis, and diverticular disease) have been reported in T2D, GERD, and Crohn's disease[50–52]. On the other hand, the organic transporter *SLC22A3* (shared by T2D and PUD) is highly expressed in the liver and intestine and associated with colorectal cancer and T2D[53,54]. Similarly, *CARD9* (shared by T2D and IBD) is a primary regulator of adaptive and innate immune functions and has been reported in T2D and IBD[55,56]. Evidence supports the role of *CARD9* in IBD (with some variants associated with increased disease risk while others have a protective effect)[55,56] and metabolic disorders (through inflammatory pathways), including obesity, insulin resistance[57], and potentially T2D[56].

Providing further insights, our pathway-based analysis robustly supports proinflammatory, immune, and particularly autoimmune mechanisms in T2D, IBD, IBS, diverticular disease, GERD, and, to a lesser extent, gastritis-duodenitis and PUD. For instance, in addition to the MHC region-linked pathways, mechanisms of autoimmune conditions were among the most overrepresented in our study. Despite its traditional classification as a purely metabolic disorder, evidence now suggests autoimmune components in the pathobiology of T2D[58,59]. Importantly, our findings suggest that excluding the MHC region does not necessarily rule out the involvement of immune or inflammatory pathways among the significantly enriched biological mechanisms shared by T2D and GI disorders. For instance, while all MHC pathways were no longer enriched in the analysis excluding the region, inflammatory or immunological mechanisms or processes were still overrepresented for genes overlapping T2D and IBD, IBS, diverticular disease, and gastritis-duodenitis. Our findings also indicate a significant enrichment of viral infection-related mechanisms, including those of viral myocarditis, Ebola virus, 'human T-cell leukaemia virus 1' and human papillomavirus, for genes overlapping T2D and GERD, IBS, diverticular disease, and IBD. There is emerging evidence supporting the mechanistic involvement of viruses in the pathology of T2D (diabetogenic virus, recently reviewed[60]) and GI disorders[61] such as IBD[62] and IBS[63,64], potentially through

systemic inflammation, viral-mediated gut microbiota perturbation and autoimmune provocation[61,65–69]. The report of several cases of new-onset diabetes[70,71] and GI disorders[61] following coronavirus SARS-CoV-2 infection (COVID-19) lends further support to this position. Viral-related mechanisms may also be involved in the risk of GERD[72].

Finally, we found unique pathways specific to certain phenotype pairs, which could suggest mechanistic differences in GI disorders or their relationship with T2D, including leptin signalling pathways for genes overlapping T2D and IBD. Leptin is recognised for its extensive modulatory roles in inflammatory and autoimmune responses[73], indicating its potential involvement in T2D and IBD, which have well-established mechanisms related to these responses. Thyroid hormone, interferon, and notch signalling pathways were also significantly enriched for genes overlapping T2D and IBS——results that are consistent with a recent study on IBS[74]—and partly, T2D and IBD. The Notch signalling pathway is crucial in regulating the differentiation and proliferation of the colonic epithelium and has also been linked to IBD (specifically, ulcerative colitis)[75]. This pathway is believed to enhance gluconeogenesis in the liver, play a role in pancreatic development[76], and contribute to hyperglycaemia[76], supporting its potential relevance in T2D[77]. We found a significant enrichment of cardiovascular-related pathways such as viral myocarditis (T2D and IBS, GERD, and diverticular disease), dilated cardiomyopathy (T2D and GERD), regulation of cardiac muscle cell action potential, abnormal QT interval and abnormal electrocardiogram (T2D and gastritis-duodenitis) suggesting a potential interplay between metabolic, gastrointestinal, and cardiovascular health. Pathways related to calcium homeostasis (such as hypercalcemia and hyperuricemia) were also overrepresented for genes overlapping T2D and PUD. While the underlying mechanisms are not fully understood, diabetes is associated with abnormalities in calcium homeostasis[78], and hypercalcemia may contribute to PUD by activating the calcium-sensing receptor in gastric parietal cells, leading to an increase in gastric acid secretion[79].

In conclusion, our study reveals significant positive global and local genetic correlations between T2D and PUD, IBS, GERD, gastritis-duodenitis, and diverticular disease. Gene-level overlap assessments reinforce these findings, indicating that T2D shares more genes with these GI disorders than expected by chance. Our analysis further reveals a putative causal influence of genetic liability to T2D on PUD, gastritis-duodenitis, and bidirectional causality with GERD, not accounted for by BMI. These results indicate a risk-increasing relationship between T2D and the named GI disorders, particularly PUD, GERD, and gastritis-duodenitis. While observational studies support increased IBD risk in T2D, our genetic-based analysis reveals a complex relationship between these disorders, implicating inverse correlations at specific genomic regions and suggesting that risk variants for one may be protective for the other. Our study highlights the importance of screening for GI disorders in patients presenting with signs and symptoms (or diagnosis) of diabetes for possible early detection, treatment, or preventative approaches where relevant. Finally, we identified several genes and biological pathways shared by T2D across GI disorders, including autoimmune, viral, and proinflammatory-mediated mechanisms, providing important targets for further investigation in these disorders.

Analysing well-powered genomic data using well-regarded statistical genetic tools is a major strength in understanding the relationship between T2D and GI disorders. This statistical genetic approach is generally less prone to biases from disease status ascertainment, small sample sizes, and residual confounding that often characterise traditional observational studies. We utilised powerful and complementary statistical genetic methods, including MR, local and global correlation approaches, and gene and pathway-based analysis—providing balanced and comprehensive evidence. Our approach is robust to many lifestyle and environmental confounders, given that it is based on genetics. However, it is essential to note that our data are predominantly from European populations, and caution should be exercised when interpreting, comparing, or generalising our findings to other ancestries. Sample overlap can confound genetic correlation or causal relationship findings. Our preliminary analysis, however, shows no substantial overlap across the GWAS of T2D and GI disorders. Additionally,

the methods and software used in our analyses contribute to making our findings reliable. For example, the LDSC method is robust to the confounding influences of many sources, including environmental factors. Findings from the method are also not biased by a possible overlap of samples, provided the genetic covariance intercepts are not constrained. We note that genetic covariance intercepts were not constrained in any of our analyses.

Similarly, we provided potential sample overlap information (LDSC-based genetic covariance intercept estimates) for our LAVA's local genetic correlation analysis, making its results highly unlikely to have been influenced by sample overlap bias. Traits analysed in our study may be linked to other factors that can contribute to or partly explain the relationship of T2D with GI disorders, and identifying or adjusting for all those is beyond the scope of our study. Nonetheless, we used BMI-adjusted data to assess the likely effects of BMI on the link between T2D and GI disorders. We acknowledge that some authors have suggested that using adjusted data, such as BMI adjustment, carries the risk of bias[80–82]. Our experience in this respect indicates a potential for over-adjustment, which may result in conservative findings based on such data. Nevertheless, we deem the analysis using this data crucial for illustrating the likely impact of BMI on the genetic relationship between T2D and GI disorders. Importantly, our results confirm genetic overlap between T2D and GI disorders, irrespective of BMI's effect. We used conservative findings or approaches in our study to mean cautious practices that prioritise robustness by using stricter measures aimed at minimising likely false positives and avoiding exaggerated claims (erring on the side of caution). This approach is appropriate but can sometimes downplay or underestimate findings.

Furthermore, we used the multivariable MR analysis to adjust for the likely impact of BMI in our causal relationship assessment, which aligns with practice in studies[35]. Additionally, we implemented several precautionary measures in our MR-based causality assessment, including strict clumping parameters and manual removal of suspected pleiotropic variants, making our causality estimates potentially conservative but also reliable and robust. Finally, identified biological pathways can sometimes be redundant, with the possibility of over-interpretation (and sometimes difficulty replicating exact pathways). However, the consistencies observed in the identified pathways and the biological functions of the shared genes provide robust support for our findings. Multiple re-runs of the analysis were consistent and did not alter the conclusion of our findings.

## Methods
### GWAS summary data for T2D and GI disorders
We analysed well-powered GWAS summary data from several publicly available repositories, including the 'DIAbetes Genetics Replication And Meta-analysis' (DIAGRAM) consortium, GWAS Atlas, the United Kingdom biobank (UKB), and other international research consortia/groups. First, we used the GWAS summary data for T2D with 74,124 cases and 824,006 controls from Mahajan et al.[83] as the primary data for analysis (Table 1 and Supplementary Data 1). These data were sourced from the DIAGRAM consortium (https://www.diagram-consortium.org, accessed November 2022). We also obtained the GWAS for BMI-adjusted T2D (74,124 cases and 824,006 controls) from the DIAGRAM consortium[83]. Additionally, we used the UKB T2D GWAS (from the Lee Lab GWAS, Phecode 250.2, cases: 18,945, controls: 388,756, https://www.leelabsg.org/resources, accessed November 2022) as the replication set for T2D (Table 1, and Supplementary Data 1) for (partial—because data may not be completely independent) replication testing.

We obtained summary data for GI disorders from many sources, including published GWAS of PUD (cases: 16,666, controls: 439,661)[43], GERD (cases: 71,522, controls: 261,079)[84], IBS (cases: 28,518, controls: 426,803)[43], and IBD (cases: 7045, controls: 449,282)[43]. The GWAS for diverticular disease (diverticulosis and diverticulitis, Phecode 562, cases: 27,311, controls: 334,783) and gastritis-duodenitis (Phecode: 535, cases: 28,941, controls: 378,124) were sourced from the UKB through the Lee Lab (https://www.leelabsg.org/resources, accessed November 2022). The

GWAS data from the Lee Lab comprise the complete subset of the White British population, primarily from the UKB. The UKB data contain information from over 500,000 individuals aged 40 years and above, collected prospectively between 2006 and 2010. GI disorders for the cohorts were surgically diagnosed, self-reported, or diagnosed according to the ICD 9 and 10 codes. Additionally, the GWAS were adjusted for genotyping batch, birth year, sex, and the first principal components.

We used additional data, for example, the BMI GWAS summary data from the UKB (phenotype field: 23104, GWAS Atlas ID: 3445, downloaded and used locally)[85], in BMI-adjusted multivariable MR analyses. These data contain a total sample of 379,831. For potential replication testing in BMI-adjusted multivariable MR analysis, we sourced another GWAS summary data for BMI from the GIANT consortium (GWAS ID: ieu-b-40, implemented online through the 'Two-SampleMR' platform) with a total sample of 681,275 individuals[45]. We also utilised T2D GWAS summary data (ebi-a-GCST006867, through the TwoSampleMR online platform), comprising 62,832 cases, 596,424 controls, and a total sample size of 659,256 individuals[44]. This T2D GWAS meta-analysed the DIAGRAM, the GERA (Genetic Epidemiology Research on Adult Health and Aging), and the UKB data. For (partial) replication testing on the side of GI disorders, we used other sets of GWAS summary statistics for GI disorders from a range of sources summarised in Supplementary Data 1. All the data utilised in the present study were from individuals of European ancestry. More comprehensive information on each of the GWAS data used in the present study, including their respective quality control procedures, have previously been documented in their separate associated publications (Table 1 and Supplementary Data 1). Analysis in this study was based on the primary data presented in Table 1 (that is, not the replication set), except where specifically noted that replication or further testing was performed.

### Genome-wide (global) genetic correlation analysis
We estimated the global genetic correlation between T2D and each GI disorder using the LDSC method[41]. LDSC is computationally efficient, user-friendly, and can uniquely distinguish actual polygenic signals from population stratification or cryptic relatedness[41]. We first converted the GWAS summary data into the LDSC format using the 'munge_sumstats.py' function. LDSC performs quality control processes, including removing duplicate and ambiguous or non-SNP variants. We estimated cross-trait bivariate global genetic correlations comparing T2D (and $T2D_{bmiadj}$) with each GI disorder. We utilised the European population LD scores computed based on the 1000 genome data and restricted our analyses to common SNPs in HapMap3.

There was no evidence of substantial sample overlap across the GWAS for T2D and GI disorders; nevertheless, we retained the genetic covariance intercepts in the present study. While our estimates may be conservative, this practice reduces the potential bias of unknown sample overlap. Finally, we tested our findings' (partial) reproducibility using a different GWAS set in assessing LDSC-based genetic correlation between T2D and GI disorders (the replication set, Table 1 and Supplementary Data 1). We considered a genetic correlation result nominally significant at $P < 0.05$ and significant at $P < 8.33 \times 10^{-3}$ (0.05/6, Bonferroni adjustment for testing six GI disorders). LDSC-based genetic correlation has values ranging from $-1$ to 1, where negative values represent divergent effects (potentially risk-decreasing) across the genome, and positive values indicate concordant results (potentially risk-increasing) of shared genetic variants. To rule out the potential impact of the MHC region on our genetic correlation estimates, we performed further analyses by excluding the MHC region (chr6: 28477797–33448354, GRCh37/hg19) from our data. The MHC region used here is based on the reference from https://www.ncbi.nlm.nih.gov/grc/human/regions/MHC?asm=GRCh37, and it has been used in several studies[86–88]. Given some authors often extend the MHC region to approximately 25–34 Mb (GRCh37/hg19)[43,89], we performed another set of analyses extending the excluded region to chr6: 25,000,000 to 34,000,000 (GRCh37/hg19).

## Local genetic correlation analysis

We conducted local genetic correlation analysis in LAVA[42], providing further insights into the relationship between T2D and GI disorders. A comprehensive description of LAVA as an analytic tool has been published[42]. Unlike the global genetic correlation analysis, which averages the genetic effect between traits across the genome, local genetic correlation tests for signals at specific genomic locations and identifies regions contributing disproportionately to the relationship between the traits[31,42]. Here, we assess the local genetic correlation of T2D (and T2D$_{bmiadj}$) with PUD, IBD, gastritis-duodenitis, IBS, GERD, and diverticular disease across the genome using LAVA and based on the locus definition file (with semi-independent LD blocks) provided in the original publication of LAVA[42]. We used the 1000 Genomes (v3) EUR[90] (MAF > 0.5%) as the reference data. Moreover, we provided genetic covariance intercept estimates (from LDSC-based bivariate genetic correlation analysis) as part of the input files for LAVA to account for potential sample overlap in local genetic correlation analyses[31,42].

Following a range of quality control procedures, including alignment of effect alleles to the reference data across all traits, LAVA first performs univariate analysis to estimate each trait's local genetic heritability (at the defined loci). Only loci with sufficient univariate signals (based on their $P$ values) progressed to bivariate local genetic correlation analysis. A user-nominated $P$ value threshold, accounting for the number of loci tested, is commonly used to decide which univariate locus has sufficient signals to progress into the bivariate analysis. However, LAVA can control for type 1 error with or without nominating a P value threshold; hence, we used a less stringent cut-off of P < 0.05 (recommended by the program developer) as a filtering step in selecting univariate signals for bivariate analysis in the present study. This approach allows more bivariate analysis to be conducted without risking false positive findings (https://github.com/josefin-werme/LAVA/issues/38). Additionally, to maximise the number of SNPs available for analysis, we performed pairwise bivariate local genetic correlation analysis one at a time for a pair of T2D and each GI disorder. LAVA extracts and utilises overlapping SNPs for analysis; hence, combining all GI disorders with T2D GWAS data in a single analysis round can substantially limit the number of SNPs available for processing, justifying the approach we used.

## Bidirectional causal relationship analysis

We assessed the potential bidirectional causal relationship between T2D and GI disorders (PUD, GERD, IBS, diverticular disease, gastritis-duodenitis, and IBD) using the MR analysis method. Underpinned by Mendel's laws of inheritance, MR mimics randomised control trials (considered the gold standard in assessing causality) and provides a cost-effective approach for a robust evaluation of the potential causal effects of exposure variables (risk factors) on outcome variables (disease outcomes)[30,91]. MR utilises genetic variants inherited at conception as instrumental variables (IVs); hence, the method is less susceptible to lifestyle confounders, underscoring why, unlike conventional observational studies, MR is less affected by limitations such as reverse causation and residual confounding[30,91]. Three primary assumptions—relevance (IVs associated with the exposure robustly), exclusion restriction (IVs influence the outcome through the exposure-outcome pathway), and independence (IVs not associated with confounders)—underlie MR analysis[30] (Fig. 3).

Here, we used the two-sample MR (2SMR) package (version 0.5.6), implemented in the R statistical platform (version 4.0.2), to assess the potential bidirectional causal association of T2D with GI disorders (PUD, IBS, gastritis-duodenitis, diverticular disease, GERD and IBD). In all analyses, we selected linkage disequilibrium (LD)-independent ($r^2 < 0.001$) SNPs strongly associated with the exposure variables (at $p < 5 \times 10^{-8}$) as IVs, with an $F$-statistic > 10[30], indicating that they are less prone to weak instrument bias. The MR package extracts the selected IVs from the outcome data, performs data harmonisation, and subsequently carries out the MR analysis. We used the inverse variance weighted (IVW) method because it provides the most reliable causal effect estimates. This method assumes no horizontal pleiotropy and is reliable, provided this underlying assumption is not violated.

In line with practice in previous studies[30,31,46,92–94], we assessed the validity of our IVW estimates using two additional models, namely, the MR-Egger (which can produce valid estimates by correcting pleiotropy even with up to 100% invalid IVs) and the weighted median (which can produce valid estimates with up to 50% invalid IVs). We also tested the assumption of no horizontal pleiotropy using the MR-Egger intercept. We infer a potential violation of this assumption wherever MR-Egger intercepts deviate significantly from zero. We performed additional analyses to test the robustness of our findings, including Cochran's $Q$ statistics for heterogeneity of SNP effects, single SNP MR, 'leave-one-out' analysis (to assess the influence of individual IVs on the overall results), and an assessment of the funnel plot for symmetry.

To further investigate the bidirectional causal relationship between T2D and each GI disorder, we used the MR-PRESSO method[95]. The method can identify potentially pleiotropic SNPs (based on its global test) and adjust for those in a corrected MR analysis (outlier test)[95]. It can also test whether the outlier SNPs biased the originally estimated causal effect in the crude MR-PRESSO analysis (distortion test)[95], providing robust and reliable results. In the present study, we first assessed the potential causal impact of T2D on each GI disorder. In a bidirectional model, we changed the direction of our analysis and investigated the causal influence of each GI disorder on T2D. We performed a multivariable Mendelian randomisation analysis using the two-sample MR package, potentially adjusting for the effect of BMI on the relationship between T2D and GI disorders. Briefly, we retrieved instruments for each exposure and then combined them into a set of all instruments. We performed clumping on these instruments, re-extracted the final clumped SNPs from each exposure and the outcome, carried out harmonisation, and subsequently performed the multivariable MR analyses on the harmonised data. We performed analysis locally (using our GWAS data) and online (using data from the OpenGWAS database), as summarised in the results section and Supplementary Data 4. We restricted the present multivariable analysis to only GI disorders that demonstrated at least a nominally significant causal association with T2D (either as exposure or outcome) in the univariable MR.

## Gene-based association analysis

We complemented our SNP-level genetic overlap analysis by extending the present study to the gene-based association level. Genes are the basic functional units of the human genome and are more closely related to biology than SNPs. Hence, gene-level analysis can provide greater power for assessing genetic overlap, identifying shared genes, and gaining insights into underlying biological mechanisms[31,46,96–99]. We first performed gene association analysis for T2D and each GI trait using the MAGMA[100] software in FUMA (an online platform, https://fuma.ctglab.nl/, accessed in November 2022). We mapped overlapping SNPs to protein-coding genes at '±0 kb outside the gene' (SNPs located within the gene). For example, in the T2D and PUD pair, we used only SNPs overlapping these two disorders in the gene-based analysis. This approach ensures that equivalent gene-based tests (having the same number of genes) were performed separately for both disorders, as in previous studies[31,46,96–98].

## Estimating gene-level genetic overlap

Here, we investigate gene-level genetic overlap by estimating the proportion of genes overlapping between T2D and each GI disorder in line with previous studies[31,46,93,96–99]. We applied a statistical test to examine whether the gene overlap proportion was more than expected by chance. For this analysis, we used the results of our gene-based analysis and assigned T2D as the discovery set and each of the GI disorder pairs as the target set. In the T2D–PUD pair (as a simplified illustration), for instance (with T2D as the discovery set and PUD as the target set), we extracted (from the MAGMA gene output files), genes associated separately with T2D and PUD at $P_{gene} < 0.05$. We first determined the number of genes represented in T2D at $P_{gene} < 0.05$ (call this 'x') and in PUD (call this 'y'). We identified how many genes overlapped between T2D and PUD at the threshold of $P_{gene} < 0.05$ (that is, how many 'x' and 'y' genes overlapped, called this 'z'). Finally, using

the binomial test, we estimated the 'expected proportion of gene overlap' and compared it with the 'observed proportion of gene overlaps'.

The 'expected proportion of gene overlap' was obtained by dividing the number of genes associated with the discovery set (T2D) at $P_{gene} < 0.05$ (that is, 'x') by the total number of genes (approximately 18,710 genes for T2D—from the MAGMA output file) in the discovery set (that is, 'x' divided by 18,710; call this 'e'). The 'observed proportion of gene overlap' was the ratio of genes overlapping T2D and PUD (that is, 'z') to the total number of genes associated with PUD (that is, 'y') at $P_{gene} < 0.05$ (meaning 'z' divided by 'y', called 'b'). Using the one-sided binomial test, we assessed whether 'b' (the 'observed proportion of gene overlap') was significantly different from 'e' (the 'expected proportion of gene overlap') [see 'result explained' in the Results section].

Where the 'observed proportion of overlap' is significantly more than the 'expected proportion of overlap' (assessed based on their $P$ values), we consider the result significant, indicating that the two traits (T2D and PUD, for example) demonstrate gene-level genetic overlap. We followed the same procedure in assessing the gene-level genetic overlap between T2D and IBS, GERD, gastritis-duodenitis, IBD, and diverticular disease. We also evaluated the relationship using the BMI-adjusted T2D data.

### Identifying genes shared by T2D and GI disorders

Following the same approach in previous studies[31,32,93,99], we used the FCP method to combine gene association $P$ values of T2D and GI disorders (T2D and PUD, for example). Subsequently, we identified their shared genes reaching genome-wide significance (GWS). In conducting this analysis, we utilised the results of equivalent gene-based association analysis (outputs from MAGMA) for the pair of T2D GWAS and each GI disorder. First, using gene association analysis output from MAGMA, we identified GWS genes separately for T2D and each GI disorder at an adjusted $P$ value of $2.62 \times 10^{-6}$ (with Bonferroni correction for testing 19,105 genes, 0.05/19105). Next, we combined the association $P$ values of T2D with the respective GI disorders using the FCP approach as in previous studies[31,32,46,93,94].). We used the results of FCP analyses to identify genes shared by T2D and each of the GI disorders under three categories—GWS genes for T2D shared by GI disorders, GWS genes for GI disorders shared by T2D, and genes not GWS in either T2D or GI disorders GWAS but which reached the status after the FCP analysis (putatively novel genes shared by T2D and GI disorders)—see the results. The T2D from Mahajan *et al.*[83] was utilised in our gene-based analysis.

### Pathway-based analysis

To make biological sense of genes shared by T2D and each of the GI disorders, as well as gain new insights into their likely underlying mechanisms, we performed pathway-based analysis using the g:GOst tool on the g-profiler platform (https://biit.cs.ut.ee/gprofiler/gost, accessed April 2023)[101]. Genes overlapping T2D and each GI disorder at $P_{gene} < 0.05$ (with significant FCP and gene-level overlap) were used for this analysis, in line with practice in previous studies[32,46,93,94]. Briefly, the g:GOst tool performs overrepresentation, gene set, or functional enrichment analysis by mapping user-inputted genes into known sources of functional information[101]. Following adjustment for multiple testing, g:GOst subsequently detects over-represented (significantly enriched) biological pathways[101].

In the present study, we followed a well-established protocol[102], restricted functional category term sizes to between 5 and 350 values, as recommended[102], and maintained the default setting for the 'advanced options.' For example, we restricted the analysis to only human-annotated genes (https://biit.cs.ut.ee/gprofiler/gost, accessed April 2023 and March 2024). Additionally, the recommended 'g: SCS algorithm' was used for correcting multiple testing, and we considered enriched biological pathways significant at the adjusted $P$ value of $P_{adjusted} < 0.05$[101,102].

### Statistical analysis and reproducibility

Our statistical analysis was primarily conducted in the Unix environment and employed the R software (https://www.r-project.org/). Additionally, we used other software tools, including Python (https://www.python.org/), Plink (https://www.cog-genomics.org/plink/), and various online platforms (G-profiler: https://biit.cs.ut.ee/gprofiler, and FUMA: https://fuma.ctglab.nl). To address the issue of multiple testing, we applied the Bonferroni method in LDSC, causality, and gene-based analyses. We implemented the 'g: SCS algorithm,' a recommended inbuilt approach, for multiple testing corrections in the G-profiler. To assess the potential reproducibility of our findings, we leveraged several GWAS data for both T2D and GI disorders.

### Ethical approval and participant consent

Our study constitutes a secondary analysis of existing GWAS summary data from public repositories and international research consortia. The specific ethics approvals for each dataset are documented in the associated publications referenced in the GWAS summary data section. No additional ethics approval is required to conduct the present study.

### Reporting summary

Further information on research design is available in the Nature Portfolio Reporting Summary linked to this article.

### Data availability

All data generated during our study are fully documented in the published article and its supplementary section. The GWAS summary statistics data that we analysed were obtained from international research consortia and public repositories, as outlined in the GWAS summary data subsection. These data are openly accessible online through the links and references in our study. Table and Supplementary Data 1 offers a comprehensive description of the data and instructions on accessing them.

### Code availability

In this study, we utilised publicly available software for our analyses. Below, we provide a list of URLs, some of which are online resources, detailed information about the software tools can be found, including, where applicable, the computer code: FUMA, G-profiler (https://biit.cs.ut.ee/gprofiler/), GWAS Catalogue (https://www.ebi.ac.uk/gwas/home), Open Target Genetics, (https://genetics.opentargets.org/), LDSC, LAVA, and Two-Sample MR.

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

## Acknowledgements

We thank the various databases and international research groups/consortia, including the UKB, for providing us with access to the GWAS data analysed in this study. Our appreciation also goes to individuals and researchers who participated in T2D and GI disorder GWASs. SML received funding from the National Health and Medical Research Council Australia (APP1161706, APP1191535), while EOA was supported by the Department of Health Western Australia, Future Health Research and Innovation—WA Near-miss Awards: Emerging Leaders Program (G1006599). This activity has been supported by the Western Australian Future Health Research and Innovation Fund. Funders had no influence on the design, analysis, or interpretation of the findings in this study.

## Author contributions

E.O.A. and S.M.L. conceived and designed the study. E.O.A. searched for and curated GWAS data, performed statistical analyses, and wrote the initial draft of the paper. T.P., E.O., S.M.L., G.V., and O.O. contributed to the writing of the paper. All authors edited and reviewed the paper. S.M.L. supervised the work.

## Competing interests

The authors declare no competing interests.
