## [Peer review file · Communications Biology]

Reviewers' comments:

Reviewer #1 (Remarks to the Author):

In the provided manuscript, the authors explore the shared genetic etiology between type 2 diabetes (T2D) and six gastrointestinal (GI) diseases/traits. The authors observed a significant positive correlation between T2D and all studied GI traits, except for inflammatory bowel disease (IBD). They also reported several local genomic regions of high correlation between T2D and GI traits.

Furthermore, the authors found a bidirectional causal relationship between T2D and gastroesophageal reflux disease (GERD). Moreover, the manuscript highlights overlapping genes and pathways between T2D with GI traits, emphasizing a role of immunological mechanisms in the development of all studied phenotypes.

While the manuscript is comprehensive and well-analyzed, there are some comments to consider:

1.) The genome-wide genetic analysis showed strong correlations between T2D and all GI traits except IBD. These correlations became significantly lower when T2D was adjusted for body mass index (BMI). The causal relationship between obesity and T2D and most of the studied GI diseases is well known. Therefore, the question arises as to whether the remaining correlations between BMI-adjusted T2D and GI diseases are still attributable to obesity? How was the BMI - on which the adjustment was based - recorded (e.g. once or several times) and to what extent does BMI fully represent obesity?

2.) The LAVA analysis showed significant local genetic correlations between T2D and GI traits. Here, loci in the MHC region were particularly prominent. The MHC region is one of the most complex in the human genome and is characterized by highly complex LD patterns. To what extent this could have contributed to false positive results? Did the authors apply LD-based clumping or similar methods to address the complexity of the MHC region and to minimize false positive results?

3.) The authors did not report Bonferroni corrected P-values for their Mendelian randomization (MR) analyses. How would this influence their results?

4.) The MHC region is also prominently represented in the gene-based and pathway analyses. In addition to the above-mentioned characteristics of the MHC region, which could have led to false positive results, the question is whether similar significant results would also be obtained if the MHC region would be excluded from the analyses ("with" and "without MHC")? To what extent would the exclusion of the MHC region support the authors' conclusions that proinflammatory, immunologic and autoimmunologic mechanisms jointly underlie T2D and GI traits?

5.) In lines 525-527, the authors state positive genetic correlations among GI traits. However, the authors did not demonstrate correlation among GI traits in the present study.

6.) In the abstract, the authors mention bidirectional causal effects of T2D on GERD, irritable bowel syndrome (IBS), PGM (medicated phenotype for GERD/ peptic ulcer disease (PUD)) and gastritis-duodenitis (GD) (lines 45-46). As far as I have understood from the manuscript, a bidirectional causal effect is only present between T2D and GERD?

Reviewer #2 (Remarks to the Author):

This is a very comprehensive and large body of work investigating the links between diabetes and gastrointestinal traits using large scale genomic data and the latest techniques for assessing genetic correlations and causality such as LD score regression, LAVA, Mendelian Randomisation. I can not identify any errors in the methodology.

Because it is such a large body of work it can be difficult to follow, it would be helpful if possible to simplify the manuscript. This could include the following:

1. In Figure 1 the GWAS sample size information could be removed as this is also included in table 1.
2. In multiple places in the results sections the results are shown in both the text in brackets and the tables. Examples include Table 2 where the results are also written out in the paragraph in page 6. Also for table 3 and the text on page 7, and throughout..

On page 6 at the end of the paragraph the authors mention that “we did not constrain the genetic covariance intercepts in the present study; therefore, our genetic correlation estimates may be conservative”. I don't completely understand this statement or what it means for the results. I think it requires further explanation.

On page 7, the results for the genetic correlations between T2D and GI disorders are shown adjusted for BMI summary stats. Which GWAS summary stats were used here? Table 1 shows two different GWAS. The strength of the genetic correlations is reduced when attempting to control for BMI. Are the authors confident that all the BMI signal is removed and this is no longer affecting the results?

Bivariate local genetic correlations pg. 8. What do the results mean? If no insight is offered from the specific loci that are identified, the details of these such as the locus numbers could be left for the supplementary files.

On the last paragraph of page 10 it is mentioned that there is a ‘suggestively’ significant causal effect of predisposition to T2D on PUD and gastritis-duodenitis. How is ‘suggestive’ defined? This is an unclear statement where the result should be stated clearly.

For those not familiar with MR analysis I think it would be better to have a table or figure summarising the results, and more simple explanation of what each of the MR tests are showing in terms of the assumptions of MR.

On page 11 can the statement “somewhat more convincing significance” be more precisely stated?

Also they mention “Relevant tests indicate no evidence of heterogeneity”, which MR tests?

The multivariable MR is used to adjust for BMI, can it also be used to test for causal associations of all the GI disorders together with T2D. As they are not all independent from each other only certain GI traits many have a causal relationship, but they are all correlated with each other.

For the results on page 15, for the Genome-wide sig genes shared by T2D and GI disorders, how much does this add to the knowledge over and above the genetic correlation results? Also the same for the FCP analysis. This is such a large volume of work, could it be reduced down, or summarised more to state only the main points.

Response to reviewers' comments

Dear Professor George Inglis. Thank you for the opportunity to address the reviewers' comments and resubmit our revised manuscript (COMMSBIO-23-4308) entitled "*Genome-wide cross-disease analyses highlight causality and shared biological pathways of type 2 diabetes with gastrointestinal disorders*" to your journal (Communications Biology). Below are our responses (in blue text) to the reviewers' comments and questions. As recommended, we provide all referee comments (unedited) and, thereafter, our point-by-point responses. Except where otherwise noted, all line numbers mentioned refer to the clean copy of our revised manuscript.

Thank you for your helpful and insightful feedback. We look forward to hearing from you soon.

Reviewers' comments:

Reviewer #1 (Remarks to the Author):

In the provided manuscript, the authors explore the shared genetic etiology between type 2 diabetes (T2D) and six gastrointestinal (GI) diseases/traits. The authors observed a significant positive correlation between T2D and all studied GI traits, except for inflammatory bowel disease (IBD). They also reported several local genomic regions of high correlation between T2D and GI traits. Furthermore, the authors found a bidirectional causal relationship between T2D and gastroesophageal reflux disease (GERD). Moreover, the manuscript highlights overlapping genes and pathways between T2D with GI traits, emphasizing a role of immunological mechanisms in the development of all studied phenotypes. While the manuscript is comprehensive and well-analyzed, there are some comments to consider:

1.) The genome-wide genetic analysis showed strong correlations between T2D and all GI traits except IBD. These correlations became significantly lower when T2D was adjusted for body mass index (BMI). The causal relationship between obesity and T2D and most of the studied GI diseases is well known. Therefore, the question arises as to whether the remaining correlations between BMI-adjusted T2D and GI diseases are still attributable to obesity? How was the BMI - on which the adjustment was based - recorded (e.g. once or several times) and to what extent does BMI fully represent obesity?

2.) The LAVA analysis showed significant local genetic correlations between T2D and GI traits. Here, loci in the MHC region were particularly prominent. The MHC region is one of the most complex in the human genome and is characterized by highly complex LD patterns. To what extent this could have contributed to false positive results? Did the authors apply LD-based clumping or similar methods to address the complexity of the MHC region and to minimize false positive results?

3.) The authors did not report Bonferroni corrected P-values for their Mendelian randomization (MR) analyses. How would this influence their results?

4.) The MHC region is also prominently represented in the gene-based and pathway analyses. In addition to the above-mentioned characteristics of the MHC region, which could have led to false positive results, the question is whether similar significant results would also be obtained

if the MHC region would be excluded from the analyses (“with” and “without MHC”)? To what extent would the exclusion of the MHC region support the authors’ conclusions that proinflammatory, immunologic and autoimmunologic mechanisms jointly underlie T2D and GI traits?

5.) In lines 525-527, the authors state positive genetic correlations among GI traits. However, the authors did not demonstrate correlation among GI traits in the present study.

6.) In the abstract, the authors mention bidirectional causal effects of T2D on GERD, irritable bowel syndrome (IBS), PGM (medicated phenotype for GERD/ peptic ulcer disease (PUD)) and gastritis-duodenitis (GD) (lines 45-46). As far as I have understood from the manuscript, a bidirectional causal effect is only present between T2D and GERD?

Reviewer #2 (Remarks to the Author):

This is a very comprehensive and large body of work investigating the links between diabetes and gastrointestinal traits using large scale genomic data and the latest techniques for assessing genetic correlations and causality such as LD score regression, LAVA, Mendelian Randomisation. I can not identify any errors in the methodology. Because it is such a large body of work it can be difficult to follow, it would be helpful if possible to simplify the manuscript. This could include the following:

1. In Figure 1 the GWAS sample size information could be removed as this is also included in table 1.

2. In multiple places in the results sections the results are shown in both the text in brackets and the tables. Examples include Table 2 where the results are also written out in the paragraph in page 6. Also for table 3 and the text on page 7, and throughout..

On page 6 at the end of the paragraph the authors mention that “we did not constrain the genetic covariance intercepts in the present study; therefore, our genetic correlation estimates may be conservative”. I don’t completely understand this statement or what it means for the results. I think it requires further explanation.

On page 7, the results for the genetic correlations between T2D and GI disorders are shown adjusted for BMI summary stats. Which GWAS summary stats were used here? Table 1 shows two different GWAS. The strength of the genetic correlations is reduced when attempting to control for BMI. Are the authors confident that all the BMI signal is removed and this is no longer affecting the results?

Bivariate local genetic correlations pg. 8. What do the results mean? If no insight is offered from the specific loci that are identified, the details of these such as the locus numbers could be left for the supplementary files.

On the last paragraph of page 10 it is mentioned that there is a ‘suggestively’ significant causal effect of predisposition to T2D on PUD and gastritis-duodenitis. How is ‘suggestive’ defined? This is an unclear statement where the result should be stated clearly.

For those not familiar with MR analysis I think it would be better to have a table or Figure summarising the results, and more simple explanation of what each of the MR tests are showing in terms of the assumptions of MR.

On page 11 can the statement “somewhat more convincing significance” be more precisely stated?

Also they mention “Relevant tests indicate no evidence of heterogeneity”, which MR tests?

The multivariable MR is used to adjust for BMI, can it also be used to test for causal associations of all the GI disorders together with T2D. As they are not all independent from each other only certain GI traits many have a causal relationship, but they are all correlated with each other.

For the results on page 15, for the Genome-wide sig genes shared by T2D and CI disorders, how much does this add to the knowledge over and above the genetic correlation results? Also the same for the FCP analysis. This is such a large volume of work, could it be reduced down, or summarised more to state only the main points.

Response to Reviewer #1

Comment: While the manuscript is comprehensive and well-analyzed, there are some comments to consider:

Response: We thank the reviewer for finding merits in our study and for the constructive feedback.

1. The genome-wide genetic analysis showed strong correlations between T2D and all GI traits except IBD. These correlations became significantly lower when T2D was adjusted for body mass index (BMI). The causal relationship between obesity and T2D and most of the studied GI diseases is well known. Therefore, the question arises as to whether the remaining correlations between BMI-adjusted T2D and GI diseases are still attributable to obesity? How was the BMI - on which the adjustment was based - recorded (e.g. once or several times) and to what extent does BMI fully represent obesity?

Response: Thank you for this comment. Although commonly used in studies (including in genetic analysis like ours¹), we agree that body mass index (BMI) may not fully represent obesity, as it does not necessarily reflect factors like fat distribution and muscle mass². Thus, after a thoughtful review, we have reconsidered our study’s use of BMI-adjusted data. We understand that using such data carries the risk of bias and the potential for ‘compromising the accuracy’ of GWAS data³⁻⁵, as also noted in a recent publication³. Importantly, we cannot be sure that the effect of obesity (BMI) was completely removed or the case that we are over-adjusting for this factor—as suggested in our ‘test analysis’. Thus, to simplify our study, enhance clarity and conciseness (recommended by the second reviewer) and avoid a likely ‘complicated approach’, we have excluded the use of GWAS data adjusted for or conditioned on BMI. Our approach aligns with practice in recent studies using similar data or methods^{1,3,6}. Also, we noted possible limitations related to this analysis in our study (lines 590 – 593). Nonetheless, we used the multivariable Mendelian randomisation (MR) method to account for

the likely effects of BMI on the causal relationship between T2D and GI disorders, which is appropriate and in line with the practice of other authors ^{1,7}.

2. The LAVA analysis showed significant local genetic correlations between T2D and GI traits. Here, loci in the MHC region were particularly prominent. The MHC region is one of the most complex in the human genome and is characterized by highly complex LD patterns. To what extent this could have contributed to false positive results? Did the authors apply LD-based clumping or similar methods to address the complexity of the MHC region and to minimize false positive results?

Response: We thank the reviewer for this comment. We agree that the major histocompatibility complex (MHC) region has complex linkage disequilibrium (LD) patterns. However, we do not expect LAVA's results (for local genetic correlation analysis) implicating the region to suffer false positive bias. In line with the example from LAVA developers, we included the MHC region (a known pleiotropic hotspot ⁸) in our analysis given the evidence supporting immunological components in our traits of interest (for example, inflammatory bowel disease ⁹). We did not perform LD clumping. However, we utilised semi-independent LD blocks (provided by the program developers) ¹⁰ for our locus definition, thus minimising LD. We now provide this information in our method section for LAVA (lines 661 – 662).

Additionally, LAVA uses several procedures for quality control processes and has inbuilt mechanisms for preventing false positives. As a filtering step, for example, the software conducts univariate analysis to estimate the local genetic heritability of traits and only loci with sufficient signals proceed to the bivariate analysis. Notably, LAVA can adjust for type 1 errors with or without specifying a P value threshold in the filtering step (<https://github.com/josefin-werme/LAVA/issues/38>), underscoring the approach we took in our study. Indeed, LAVA developers assert that 'via extensive simulations, we show that LAVA produces unbiased estimates with well-controlled type 1 error rates, with superior performance compared to existing approaches' (pg. 279)¹⁰. Lastly, LAVA's results implicating the MHC region were extensively replicated in the gene-based approach, providing additional support and suggesting that false positive findings are unlikely.

3. The authors did not report Bonferroni corrected P-values for their Mendelian randomization (MR) analyses. How would this influence their results?

Response: We thank the reviewer for this comment. We have made the correction, which reads as follows: 'We consider the results of our MR analysis significant at $P < 8.33 \times 10^{-3}$ (0.05/6, Bonferroni adjustment for testing six GI disorders) and nominally significant at $P < 0.05$ ' (lines 268 – 270).

4. The MHC region is also prominently represented in the gene-based and pathway analyses. In addition to the above-mentioned characteristics of the MHC region, which could have led to false positive results, the question is whether similar significant results would also be obtained if the MHC region would be excluded from the analyses ("with" and "without MHC")? To what extent would the exclusion of the MHC region support the authors' conclusions that proinflammatory, immunologic and autoimmunologic mechanisms jointly underlie T2D and GI traits?

Response: We believe that achieving a consistent result across the methods strengthens the evidence and contributes to the overall confidence in our findings. Briefly, the evidence for the

MHC region cut across several aspects of our gene-based analysis, supporting the LAVA-based local genetic correlation findings. For example, the MHC region is well represented among genes already genome-wide significant (sentinel) for T2D and many GI disorders (Supplementary Tables 5 – 11), suggesting it is known for some of these traits. Assessment of shared sentinel genes reiterates this pattern (Supplementary Tables 12 – 21) with further support from others, particularly Fisher’s combined P-value analysis (Supplementary Tables 28 – 33). In line with our pathway-based analysis, the number of MHC genes for or those shared with T2D was less for peptic ulcer disease and gastritis-duodenitis. The level of consistency between the local genetic correlation, gene, and pathway-based analyses (concerning the MHC region) is more than expected by chance and so unlikely to be false positive results.

Moreover, we found other loci shared by T2D and GI disorders outside the MHC region. Excluding the MHC region, thus, means only those regions would be prominent in our analyses. Performing separate analyses for the ‘with’ or ‘without MHC’ region should not affect the significance of the identified ‘non-MHC’ loci, genes, or pathways. However, given the concern expressed by the reviewer, we carried out further assessment as follows:

- 1) We rerun our LD score regression analysis using data with and without the MHC region to rule out the possibility that the MHC region primarily drives the global genetic correlations observed between these traits. This additional analysis confirms the significant genetic correlation between T2D and GI disorders not driven by the MHC region—with little or no difference in the estimates and significance (Supplementary Table 2). We summarised this result in lines 185 – 186. Unlike the local correlation and gene-based approaches (focusing on specific regions), global genetic correlation captures the overall shared genetics across the genome. Thus, our follow-up analysis highlights the influence of other shared regions and is consistent with findings of local correlation and gene-based methods implicating some non-MHC regions/genes.
- 2) To address the question ‘To what extent would the exclusion of the MHC region support the authors’ conclusions that proinflammatory, immunologic and autoimmunologic mechanisms jointly underlie T2D and GI traits?’, we excluded the MHC region from the genes utilised in our pathway-based analysis (using T2D-IBD as an example). As expected, all MHC-related pathways were no longer observed (Supplementary Table 45). However, other inflammatory and immune-related pathways, including the ‘Th1 and Th2 cell differentiation (KEGG: 04658), ‘Th17 cell differentiation’ (KEGG: 04659), leptin signalling pathway (WP: WP2034), and Notch signalling pathway (WP: WP61) (Supplementary Table 45), etc., remained, further supporting our conclusion. In addition, many of the shared non-MHC genes play roles in inflammatory and immune-related mechanisms.

5. In lines 525-527, the authors state positive genetic correlations among GI traits. However, the authors did not demonstrate correlation among GI traits in the present study.

Response: Thank you for this observation. We have included the results for this statement in Supplementary Table 2.

6. In the abstract, the authors mention bidirectional causal effects of T2D on GERD, irritable bowel syndrome (IBS), PGM (medicated phenotype for GERD/ peptic ulcer disease (PUD)) and gastritis-duodenitis (GD) (lines 45-46). As far as I have understood from the manuscript,

a bidirectional causal effect is only present between T2D and GERD?

Response: Thank you for bringing this to our attention. We have made the necessary correction: “Univariable and multivariable Mendelian randomisation suggests causal effects of T2D on PUD and gastritis-duodenitis and bidirectionally with GERD” (lines 44 – 45).

Thank you again for your helpful comments.

Response to Reviewer #2

This is a very comprehensive and large body of work investigating the links between diabetes and gastrointestinal traits using large scale genomic data and the latest techniques for assessing genetic correlations and causality such as LD score regression, LAVA, Mendelian Randomisation. I cannot identify any errors in the methodology.

Response: Thank you for the positive feedback and for finding merit in our study. We appreciate it.

Because it is such a large body of work it can be difficult to follow, it would be helpful if possible to simplify the manuscript. This could include the following:

Response: We have made corrections to simplify this manuscript as much as possible for clarity and ease of readers’ comprehension. For instance, we removed the data adjustment analyses (BMI adjustment) to simplify our work and avoid potential bias. This modification reduced the information in various sections of our revised manuscript, including methods, results description, and Tables. Also, to make our work more reader-friendly, we have limited our MR analysis to traditional and multivariable approaches, which provide replicable and concise findings. As a result, we removed information related to other MR approaches, including in-text (methods and results sections), Supplemental notes, tables, and plots, which made our work dense. Our approach aligns with a similar study¹ and is justified by the absence of controversial results warranting further testing. To improve the presentation and focus on shared genes, we removed Table 7, which primarily displayed a subset of shared genes, mainly within the MHC region, and does not fully represent our findings. This information is now provided in detail within the relevant Supplementary Tables, allowing us to minimise redundancy in the main text. Enrichment mapping, auto-annotation analysis, and Figure 4 (in the original submission) have also been excluded. Our pathway-based findings have been summarised in the text, with comprehensive details in Supplementary Tables. Eliminating the Figure and associated analyses has been done with necessary considerations to maintain our study’s coherence and clarity.

1. In Figure 1 the GWAS sample size information could be removed as this is also included in table 1.

Response: We have removed the information about sample size from Figure 1 (shown below).

Figure 1: Workflow summarising the study design and methods used in this study

GI: gastrointestinal, GWAS: genome-wide association studies, FCP: Fisher’s combined P value, 2SMR: two-sample Mendelian randomisation, LAVA: local analysis of [co]variant association, LDSC: linkage disequilibrium score regression, T2D: type 2 diabetes. Figure created using Lucidchart (<https://lucid.app>)¹¹.

2. In multiple places in the results sections the results are shown in both the text in brackets and the tables. Examples include Table 2 where the results are also written out in the paragraph in page 6. Also for table 3 and the text on page 7, and throughout.

We thank the reviewer for this comment. We have made the necessary corrections and removed Table 3 and its associated text. However, we retained some text for parts of Table 2 (and other relevant Tables) to summarise and interpret key findings. We believe this is crucial and aligns with practice in related studies^{1,12}. and recommendations from reviewers in our past publications using similar methods. As per the reviewer’s recommendation, we did not describe the remaining results in-text for Table 2 (the replication testing part) and as applicable in other relevant Tables.

3. On page 6 at the end of the paragraph the authors mention that “we did not constrain the genetic covariance intercepts in the present study; therefore, our genetic correlation estimates may be conservative”. I don’t completely understand this statement or what it means for the results. I think it requires further explanation.

Response: LDSC uses the genetic covariance intercept to protect against bias from sample overlap while estimating genetic correlation¹³. However, if we are confident that there are no sample overlaps, constraining the intercept can reduce the standard error with the potential for making findings more significant. So, by constraining the intercept (when appropriate), we can

improve the significance of our results. In the current study, however, we did not constrain the intercept even though our initial check shows no evidence of significant sample overlap between the T2D and GI GWAS data we analysed; hence, we consider our approach conservative. This information and rationale supporting our approach is summarised in the methods section (lines 646 - 648).

4. On page 7, the results for the genetic correlations between T2D and GI disorders are shown adjusted for BMI summary stats. Which GWAS summary stats were used here? Table 1 shows two different GWAS. The strength of the genetic correlations is reduced when attempting to control for BMI. Are the authors confident that all the BMI signal is removed and this is no longer affecting the results?

Response: Following the use of the BMI-adjusted data (from Mahajan *et al.* 2018¹⁴ T2D GWAS), there was a notable reduction in the strength and significance of our correlation analyses. Given comments from this and the first reviewer, we have reconsidered our study's use of BMI-adjusted data after a thoughtful review. Using adjusted data (BMI-adjusted T2D in this case) is noted to carry a risk of bias³⁻⁵. In addition, we cannot be sure that all BMI signals were removed. We could also have over-adjusted (as our 'test' analysis suggests). Thus, we have simplified our study while enhancing clarity by simply following the approach in previous studies, that is, without using GWAS data conditioned or adjusted for BMI^{1,3,6}. We used the multivariable MR to account for the likely effect of BMI on the causal relationship, which is appropriate and aligns with previous studies^{1,7}. Also, we mentioned potential limitations concerning this analysis in our research (lines 590 – 593).

5. Bivariate local genetic correlations pg. 8. What do the results mean? If no insight is offered from the specific loci that are identified, the details of these such as the locus numbers could be left for the supplementary files.

Response: If the concern here relates to the locus number (for example, 961, 963, etc.), the sentence that followed (lines 207 – 211) clarifies what the locus numbers represent. Also, we described the local genetic correlation results and provided further insights into the findings in the discussion section. We believe that insights have been provided for the results.

6. On the last paragraph of page 10 it is mentioned that there is a 'suggestively' significant causal effect of predisposition to T2D on PUD and gastritis-duodenitis. How is 'suggestive' defined? This is an unclear statement where the result should be stated clearly.

Response: We appreciate the reviewer for this observation. We have made a correction which reads: 'We consider the results of our MR analysis significant at $P < 8.33 \times 10^{-3}$ (0.05/6, Bonferroni adjustment for testing six GI disorders) and nominally significant at $P < 0.05$ ' (lines 268 – 270). Consequently, we have replaced 'suggestively' with 'nominally'.

7. For those not familiar with MR analysis I think it would be better to have a table or Figure summarising the results, and more simple explanation of what each of the MR tests are showing in terms of the assumptions of MR.

Response: We appreciate this comment and have provided a Figure summarising MR assumption (now Figure 3 shown below).

Figure 3: Mendelian randomisation overview and assumptions

The Figure illustrates Mendelian randomisation and its three underlying assumptions¹⁵: 1) Genetic variants (SNPs) used as instrumental variables are robustly associated with the exposure, 2) SNPs for the exposure variables are not associated with confounders, and 3) SNPs for the exposure variables influence the outcomes through no other pathway but the exposure. In the first round of analysis, we utilised GI disorders as exposure variables and T2D as the outcome (shown in the Table). In the reverse analysis, we used T2D as the exposure and GI disorders as the outcome variables. GI: gastrointestinal, SNP: single nucleotide polymorphism, T2D: type 2 diabetes.

Also, we have provided another in-text Figure to concisely summarise parts of the MR results (now Figure 4, shown below). Supplementary Table 4 presents more detailed results for our MR analysis.

Exposure	Outcome	Method	OR (95% CI)	nIV	[P]	P#
T2D	GERD	IVW	1.03 (1.01 to 1.04)	143	0.29	3.80×10^{-3}
		Weighted median	1.03 (0.99 to 1.06)	143	0.29	0.1
		MR Egger	1.01 (0.96 to 1.03)	143	0.29	0.67
		MR-PRESSO	1.02 (1.01 to 1.04)	143	0.29	4.05×10^{-3}
	PUD	IVW	1.03 (1.01 to 1.07)	159	0.63	3.29×10^{-2}
		Weighted median	1.07 (1.02 to 1.14)	159	0.63	1.34×10^{-2}
		MR Egger	1.04 (0.98 to 1.13)	159	0.63	0.17
		MR-PRESSO	1.04 (1.01 to 1.07)	159	0.63	2.20×10^{-2}
	Gastritis-duodenitis	IVW	1.03 (1.01 to 1.04)	172	0.10	3.28×10^{-2}
		Weighted median	1.03 (0.99 to 1.08)	172	0.10	0.19
		MR Egger	0.99 (0.94 to 1.03)	172	0.10	0.62
		MR-PRESSO	1.03 (1.00 to 1.05)	172	0.10	2.67×10^{-2}
GERD	T2D	IVW	1.19 (1.03 to 1.37)	19	0.37	1.39×10^{-2}
		Weighted median	1.15 (1.01 to 1.30)	19	0.37	3.45×10^{-2}
		MR Egger	0.79 (0.33 to 1.92)	19	0.37	0.61
		MR-PRESSO	1.11 (0.99 to 1.23)	19	0.37	0.09

← 0.5 1.0 2.0 →
Odds ratio

Figure 4: Putative bidirectional causality between type 2 diabetes and gastrointestinal disorders

MR: Mendelian randomisation, T2D: type 2 diabetes, PUD: peptic ulcer disease, GERD: gastroesophageal reflux disease, IVW, inverse variance weighted, MR-PRESSO: Mendelian randomisation pleiotropy residual sum and outlier, P: P-value, OR: odds ratio, CI: confidence interval, nIV: number of instrumental variables, [P]: MR-Egger intercept p-value, # The p-

value for each of the model. Note: 1) We reported the corrected estimates in MR-PRESSO and crude estimates where there are no corrected results. 2) We used strict clumping parameters and manual removal of potentially pleiotropic SNPs (see methods) in the present study, which may mean a more conservative approach.

8. On page 11 can the statement “somewhat more convincing significance” be more precisely stated?

Response: We have removed this statement.

9. Also they mention “Relevant tests indicate no evidence of heterogeneity”, which MR tests?

Response: We have now mentioned the tests: MR-Egger intercepts (for horizontal pleiotropy) and Cochran’s Q statistics (for heterogeneity) [see lines 284 – 285].

10. The multivariable MR is used to adjust for BMI, can it also be used to test for causal associations of all the GI disorders together with T2D. As they are not all independent from each other only certain GI traits many have a causal relationship, but they are all correlated with each other.

Response: We appreciate the merit of this comment and gave it a consideration, but the direction of analysis does not lend itself to this suggestion. T2D (as an exposure variable) was significantly (or nominally) associated with three GI disorders (as outcome variables). In contrast, only one of the GI disorders (GERD) showed nominally significant association as an exposure variable with T2D as the outcome variable. Hence, performing multivariable MR analysis adjusting for other GI disorders that were not significant (or at least nominally significant) in their association with T2D is not indicated here. Consequently, we limit the present assessment to the likely effect of BMI on the relationship of T2D with GI disorders, which is consistent with a previous study ¹.

11. For the results on page 15, for the Genome-wide sig genes shared by T2D and CI disorders, how much does this add to the knowledge over and above the genetic correlation results? Also the same for the FCP analysis.

Response: Genetic correlation (global) is a measure that averages the genetic effect between traits across the genome. We can gain insights into the potential shared genetic relationships between traits by estimating the global genetic correlation. However, this information is insufficient to identify the specific loci associated with the relationship between the characteristics. Therefore, we carried out further analyses, such as the local genetic correlation and gene-level analyses. Moreover, the gene-level analysis assessed the overlap between genes, providing greater power for analysis than SNP-level correlation assessments. This analysis also enabled us to identify shared genes that play a role in the biological mechanisms of the traits assessed. In addition, identifying genes that reach genome-wide significance, either the sentinel or through the FCP approach provides valuable insights into the underlying biological mechanisms of the traits beyond what is possible through genetic correlation analysis alone. Each analysis method has its aim and importance, summarised in lines 146-161 and elaborated upon in the methods section.

12. This is such a large volume of work, could it be reduced down, or summarised more to state only the main points.

Response: We agree that our study is extensive, and to make it more precise and concise, we have removed certain aspects as recommended by the reviewer. Specifically, we have excluded data adjustment analysis (BMI adjustment, as explained above) to simplify our work and make it more accessible. This step substantially reduced the volume of information presented in our study, such as in the methods, results description, Table 2, parts of Table 6 (in the original manuscript submitted), and so on. We have also limited our MR analysis to traditional and multivariable approaches, which provide replicable and concise findings. As a result, we have removed information related to other MR approaches, including in-text (methods and results sections), Supplemental notes, tables, and plots, that may have made our work dense.

Additionally, we have removed Table 7, as it only shows a few shared genes (mainly in the MHC region) and does not fully represent the shared genes we identified. The relevant supplementary Tables provide this information in detail, allowing us to reduce the volume of information presented in-text. Furthermore, we have removed enrichment mapping and auto-annotation analysis, along with Figure 4 (in the original submission). Our pathway-based findings have already been summarised in-text with detailed information in Supplementary Tables. Thus, we believe that removing the Figure and its associated analyses will not affect the overall presentation of our study. Lastly, we effect changes where appropriate with the view to make our work concise and easy to follow.

Thank you again for your helpful comments.

References

- 1 Chen, J. *et al.* Gastrointestinal Consequences of Type 2 Diabetes Mellitus and Impaired Glycemic Homeostasis: A Mendelian Randomization Study. *Diabetes Care* **46**, 828-835, doi:10.2337/dc22-1385 (2023).
- 2 Nuttall, F. Q. Body Mass Index: Obesity, BMI, and Health: A Critical Review. *Nutrition Today* **50**, 117-128, doi:10.1097/nt.0000000000000092 (2015).
- 3 Niu, Y.-y., Aierken, A. & Feng, L. Unraveling the link between dietary factors and cardiovascular metabolic diseases: Insights from a two-sample Mendelian Randomization investigation. *Heart & Lung* **63**, 72-77 (2024).
- 4 Walker, V. M. *et al.* The consequences of adjustment, correction and selection in genome-wide association studies used for two-sample Mendelian randomization. *medRxiv*, 2020.2007.2013.20152413 (2020).
- 5 Hartwig, F. P., Tilling, K., Davey Smith, G., Lawlor, D. A. & Borges, M. C. Bias in two-sample Mendelian randomization when using heritable covariable-adjusted summary associations. *International Journal of Epidemiology* **50**, 1639-1650, doi:10.1093/ije/dyaa266 (2021).
- 6 Cao, H., Baranova, A., Wei, X., Wang, C. & Zhang, F. Bidirectional causal associations between type 2 diabetes and COVID-19. *Journal of Medical Virology* **95**, e28100 (2023).
- 7 Joe, G., Maria Carolina, B., George Davey, S. & Eleanor, S. Multivariable MR can mitigate bias in two-sample MR using covariable-adjusted summary associations. *medRxiv*, 2022.2007.2019.22277803, doi:10.1101/2022.07.19.22277803 (2022).
- 8 Watanabe, K. *et al.* A global overview of pleiotropy and genetic architecture in complex traits. *Nature genetics* **51**, 1339-1348 (2019).
- 9 Silva, F. A., Rodrigues, B. L., Ayrizono, M. L. & Leal, R. F. The Immunological Basis of Inflammatory Bowel Disease. *Gastroenterol Res Pract* **2016**, 2097274, doi:10.1155/2016/2097274 (2016).
- 10 Werme, J., van der Sluis, S., Posthuma, D. & de Leeuw, C. A. An integrated framework for local genetic correlation analysis. *Nature Genetics* **54**, 274-282, doi:10.1038/s41588-022-01017-y (2022).
- 11 Faulkner, A. & Contributor. Lucidchart for easy workflow mapping. *Serials Review* **44**, 157-162 (2018).
- 12 Adewuyi, E. O. *et al.* Genetic analysis of endometriosis and depression identifies shared loci and implicates causal links with gastric mucosa abnormality. *Human Genetics* **140**, 529–552, doi:10.1007/s00439-020-02223-6 (2021).
- 13 Bulik-Sullivan, B. K. *et al.* LD Score regression distinguishes confounding from polygenicity in genome-wide association studies. *Nature genetics* **47**, 291-295, doi:10.1038/ng.3211 (2015).
- 14 Mahajan, A. *et al.* Fine-mapping type 2 diabetes loci to single-variant resolution using high-density imputation and islet-specific epigenome maps. *Nature Genetics* **50**, 1505-1513, doi:10.1038/s41588-018-0241-6 (2018).
- 15 Davies, N. M., Holmes, M. V. & Davey Smith, G. Reading Mendelian randomisation studies: a guide, glossary, and checklist for clinicians. *BMJ* **362**, k601, doi:10.1136/bmj.k601 (2018).

Reviewers' comments:

Reviewer #1 (Remarks to the Author):

I am still not convinced with the revised version of the manuscript by Adewuyi et al. ("Genome-wide cross-disease analyses highlight causality and shared biological pathways of type 2 diabetes with gastrointestinal disorders") and the reply to my comments. In the following, I will address my points of concern, retaining the numbering of my first comments:

ad 1) In the previous version of the manuscript, the correlations of T2D and GI traits were also shown adjusted for BMI. As result, they were significantly lower. In the revised version of the manuscript, the authors did not follow my suggestion to address this. Instead, they removed the BMI-adjusted data in the current manuscript version and justify this in the discussion by claiming that it is outside the scope of their work. I have a different opinion, as potential readers of the manuscript are now unable to assess how BMI influences their results and conclusion (see title of the manuscript).

ad 2 and 4) My comment was directed at how the inclusion of the associations in the MHC region might have influenced the results of the correlation and pathway analyses. The authors now present the results of the correlation analyses excluding the MHC region in ST2. The results using the T2D GWAS of Mahajan et al. (2018) are identical to the results including the MHC region. This simply is not possible. With regard to the pathway analyses, the authors show, using IBD as example, that the exclusion of the MHC region (ST45) confirms their conclusion that immunological processes jointly underlie T2D and GI traits. This is not surprising, as IBD is an autoimmune disease. By looking at the pathway data for the other GI traits, the influence of the MHC region becomes obvious (ST40-44). I do not believe that immunological processes would be indicated in these traits if the MHC region would be removed from the analyses.

Reviewer #2 (Remarks to the Author):

The authors have responded fully and comprehensively to all the reviewers comments, and have improved the clarity of the manuscript.

Response to reviewer's comments

Dear Reviewer

We appreciate the time and effort you dedicated to providing helpful comments and constructive feedback on our manuscript, COMMSBIO-23-4308A, entitled “*Genome-wide cross-disease analyses highlight causality and shared biological pathways of type 2 diabetes with gastrointestinal disorders*”. We have thoroughly considered all the comments and revised our manuscript accordingly or provided rebuttal as appropriate. Below are our detailed responses (in blue text). As recommended, we provide all referee comments (unedited) and our point-by-point responses to Reviewer 1’s concerns, as we had previously addressed all concerns to the satisfaction of Reviewer 2.

Reviewers’ comments:

Reviewer #1 (Remarks to the Author):

I am still not convinced with the revised version of the manuscript by Adewuyi et al. (“Genome-wide cross-disease analyses highlight causality and shared biological pathways of type 2 diabetes with gastrointestinal disorders”) and the reply to my comments. In the following, I will address my points of concern, retaining the numbering of my first comments:

ad 1) In the previous version of the manuscript, the correlations of T2D and GI traits were also shown adjusted for BMI. As result, they were significantly lower. In the revised version of the manuscript, the authors did not follow my suggestion to address this. Instead, they removed the BMI-adjusted data in the current manuscript version and justify this in the discussion by claiming that it is outside the scope of their work. I have a different opinion, as potential readers of the manuscript are now unable to assess how BMI influences their results and conclusion (see title of the manuscript).

ad 2 and 4) My comment was directed at how the inclusion of the associations in the MHC region might have influenced the results of the correlation and pathway analyses. The authors now present the results of the correlation analyses excluding the MHC region in ST2. The results using the T2D GWAS of Mahajan et al. (2018) are identical to the results including the MHC region. This simply is not possible. With regard to the pathway analyses, the authors show, using IBD as example, that the exclusion of the MHC region (ST45) confirms their conclusion that immunological processes jointly underlie T2D and GI traits. This is not surprising, as IBD is an autoimmune disease. By looking at the pathway data for the other GI traits, the influence of the MHC region becomes obvious (ST40-44). I do not believe that immunological processes would be indicated in these traits if the MHC region would be removed from the analyses.

Reviewer #2 (Remarks to the Author):

The authors have responded fully and comprehensively to all the reviewers comments, and have improved the clarity of the manuscript.

Our responses are provided below

Reviewer #1 (Remarks to the Author):

I will address my points of concern, retaining the numbering of my first comments:

Comment 1: ad 1) In the previous version of the manuscript, the correlations of T2D and GI traits were also shown adjusted for BMI. As result, they were significantly lower. In the revised version of the manuscript, the authors did not follow my suggestion to address this. Instead, they removed the BMI-adjusted data in the current manuscript version and justify this in the discussion by claiming that it is outside the scope of their work. I have a different opinion, as potential readers of the manuscript are now unable to assess how BMI influences their results and conclusion (see title of the manuscript).

Response to comment 1

In the previous version, the analysis using the BMI-adjusted type 2 diabetes (T2D) data was primarily removed to address the potential risk of bias (from our literature review) and evidence from our ‘test analysis’ which suggests a possible over-adjustment (that is, potentially conservative) and to address the second reviewer’s comments to streamline the analysis and provide a more concise manuscript.

However, to address the current concern of the reviewer, we have done the following:

A. We reintroduce our analyses between BMI-adjusted T2D and gastrointestinal (GI) disorders for:

- (i) Global genetic correlation using the LDSC method. The new addition for this analysis in-text reads thus:

“Furthermore, we investigated the genetic correlation between T2D and GI disorders using BMI-adjusted T2D GWAS summary data ($T2D_{bmiadj}$) to account for the potential effect of BMI on the relationship between these disorders. Our findings reveal a reduction in the strength of the genetic correlation between $T2D_{bmiadj}$ and GI disorders. However, T2D maintains a positive and significant genetic correlation with all GI disorders except IBD. We found a significant genetic correlation of $T2D_{bmiadj}$ with PUD ($r_g = 0.19$, $se = 0.05$, $P = 2.92 \times 10^{-5}$), GERD ($r_g = 0.24$, $se = 0.03$, $P = 5.11 \times 10^{-20}$), gastritis-duodenitis ($r_g = 0.25$, $se = 0.04$, $P = 5.23 \times 10^{-9}$), IBS ($r_g = 0.10$, $se = 0.04$, $P = 7.88 \times 10^{-3}$), PGM ($r_g = 0.25$, $se = 0.04$, $P = 4.29 \times 10^{-22}$) and diverticular disease ($r_g = 0.11$, $se = 0.03$, $P = 6.13 \times 10^{-4}$) [Supplementary Table 2]. The genetic correlation between T2D and IBD remained nonsignificant in the BMI-adjusted analysis ($r_g = 0.02$, $se = 0.05$, $P = 6.98 \times 10^{-1}$). We replicated these findings using additional GWAS data for GI disorders (Supplementary Table 2) [lines 203 – 212].

- (ii) Local genetic correlation using the LAVA method. The new addition for this analysis reads:

“Additionally, most loci identified were replicated in the BMI-adjusted analysis (using the $T2D_{bmiadj}$ GWAS, Supplementary Table 3), providing evidence that BMI does not account for the identified loci shared by T2D and GI disorders.” (Lines 230 – 232).

- (iii) Gene-based genetic overlap analysis: The new addition for this analysis reads thus:

“To explore the potential contribution of BMI to this relationship, we further assessed gene-level genetic overlap using T2D data adjusted for BMI. Findings from this analysis reveal a

slight reduction in the significance of our estimates; nonetheless, T2D maintains a statistically significant genetic overlap with all GI disorders (Table 4).” [Lines 376 – 379].

- (iv) Identification of genome-wide significant (GWS) genes for T2D and T2Dbmiadj. The new addition reads:

“Our gene association analysis identified genome-wide significant (GWS) genes ($P_{gene} < 2.62 \times 10^{-6}$, that is, genes that were already GWS in our data, ‘sentinel genes’) with T2D having the highest number at 504 ($P_{gene-T2D} < 2.62 \times 10^{-6}$, Supplementary Table 5), and 307 for T2D adjusted for BMI, revealing less number of SNPs when BMI was accounted for (Supplementary Table 5).” [Lines 381 – 384].

- (v) Identification of GWS genes shared by T2Dbmiadj and GI disorders. The new addition reads:

“Using the BMI-adjusted T2D, the number of overlapping GWS genes decreased slightly; for example, we found nine (TNXB, ATF6B, C6orf10, HLA-DRA, HLA-DQA1, HLA-DQB1, CARD9, EHMT2, and HLA-DRB1) sentinel genes ($P_{gene-T2D-bmiadj} < 2.62 \times 10^{-6}$ and $P_{gene-GI-disorders} < 2.62 \times 10^{-6}$) shared by both T2Dbmiadj and IBD (Supplementary Table 12), three (AGER, EHMT2, and HLA-DRB1) by T2Dbmiadj and diverticular disease (Supplementary Table 13), two (MPHOSPH9, and PITPNM2) by T2Dbmiadj and IBS (Supplementary Table 14), one (TCF4) by T2Dbmiadj and GERD (Supplementary Table 14), and one (SLC22A3) by T2Dbmiadj and PUD (Supplementary Table 14).” [Lines 401 – 407]

- (vi) Identification of shared genes reaching genome-wide significance for T2D and GI disorders according to three categories:

In the first category, that is, GWS T2D ($P_{gene-T2D} < 2.62 \times 10^{-6}$) genes that were also associated with GI disorders (at $P_{gene-GI-disorder} < 0.05$), the new addition reads:

“Using the T2D adjusted for BMI, the number of overlapping genes was slightly lower. For example, we found 30 GWS genes for T2Dbmiadj ($P_{gene-T2D-bmiadj} < 2.62 \times 10^{-6}$) that were also associated with PUD ($P_{gene-PUD} < 0.05$) [Supplementary Table 16], 105 associated with GERD ($P_{gene-GERD} < 0.05$) [Supplementary Table 17], 63 with gastritis-duodenitis ($P_{gene-gastritis-duodenitis} < 0.05$) [Supplementary Table 18], 90 with diverticular disease ($P_{gene-diverticular-disease} < 0.05$) [Supplementary Table 19], 56 with IBS ($P_{gene-IBS} < 0.05$) [Supplementary Table 20] and 65 with IBD ($P_{gene-IBD} < 2.62 \times 10^{-6}$) [Supplementary Table 21].” (Lines 421 – 427).

In the second category, that is, GWS GI disorders ($P_{gene-GI-disorder} < 2.62 \times 10^{-6}$) genes that were similarly associated with T2D (at $P_{gene-T2D} < 0.05$), the new addition reads:

“Using the T2D adjusted for BMI, we observed a slight reduction in the number of overlapping genes or their significance as documented in Supplementary Tables 22 – 27.” (Lines 434 – 436).

In the third category, that is, nominally significant genes that were not GWS in T2D ($0.05 < P_{gene-T2D} < 2.62 \times 10^{-6}$) or in GI disorders ($0.05 < P_{gene-GI-disorder} < 2.62 \times 10^{-6}$) but reached GWS status after the FCP analysis ($P_{FCP} < 2.62 \times 10^{-6}$), the new addition reads:

“In the analysis with T2Dbmiadj data, we observed a slight reduction in the number and significance of overlapping genes as shown in Supplementary Tables 28-33.” (Lines 443 – 445).

- (vii) *Pathway-based analysis: We used genes overlapping BMI-adjusted T2D and GI disorders in performing this analysis to potentially rule out the likely influence of BMI on the identified*

shared biological pathways/mechanisms or processes. To address the concern about the MHC region, we conducted this analysis with and without the MHC region (details in comments 2 below). The genes utilised for the analyses are presented in Supplementary Tables 40 – 51). The results of the pathways are presented in Supplementary Tables 52 – 57.

- B. Given that some authors have noted the likely bias associated with using adjusted data and considering our own experience, which suggests possible conservative findings, we have acknowledged this caveat in the limitation sub-section of the manuscript. The new addition reads as follows:

“We acknowledge that some authors have suggested that using adjusted data, such as BMI adjustment, carries the risk of bias¹⁻³. Our experience in this respect indicates a potential for over-adjustment, which may result in conservative findings based on such data. Nevertheless, we deem the analysis using this data crucial for illustrating the likely impact of BMI on the genetic relationship between T2D and GI disorders. Importantly, our results confirm genetic overlap between T2D and GI disorders, irrespective of BMI’s effect. We used conservative findings or approaches in our study to mean cautious approaches that prioritise robustness by using stricter measures aimed at minimising likely false positives and avoiding exaggerated claims (erring on the side of caution). This approach is not bad but can sometimes downplay or under-estimate findings” (Lines 680 – 686).

- C. The second reviewer had earlier requested that we make our study concise, and we had considered this feedback in the first revision, partly contributing to the decision to exclude the BMI-adjusted analysis. As we reintroduce this analysis in the current revision, we ensured that extensive results are provided in the Supplementary section, not in-text, to maintain the conciseness requested by Reviewer 2. Specifically:

- We present the results of the global genetic correlation analysis and reproducibility in Supplementary Table 2.
- We present the LAVA’s local genetic correlation analysis results in Supplementary Table 3.
- The results for the gene-level genetic overlap are not large; hence, we included them in Table 4
- The results for shared genes are large and varied; hence, we present those in Supplementary Tables described in section A, above.
- We summarised findings regarding the additional analyses in-text (as highlighted in section A above), thereby maintaining the conciseness of our manuscript as much as possible.

Based on the findings from these analyses, we note as follows:

1. The analysis based on BMI-adjusted data reveals significant genetic correlation and gene-level overlap between T2D and GI disorders. However, estimates and significance status were slightly lower compared to the assessment using the unadjusted data. The findings suggest that BMI partly contributes to the genetic correlation between T2D and GI disorders; however, a genetic relationship exists between these disorders that is not accounted for by BMI.
2. The number of shared genes reduced slightly for analysis using the BMI-adjusted T2D data. While this may be conservative, it suggests the likely effect of BMI was accounted for, enabling better insights into the relationship between these disorders independent of BMI.

3. Furthermore, we used the MR approach to assess the likely effect of BMI on the potential causal association between T2D and GI disorders, which was consistent with previous studies and our present objective.
4. We have corrected the relevant sections of our methods and discussion to reflect this revision. For example, alongside the MR analysis results, we corrected our discussion section, which now reads:

“We assessed the potential influence of BMI on the genetic relationship between T2D and GI disorders using analysis based on BMI-adjusted T2D data and statistical adjustment using the MR analysis method. In the BMI-adjusted T2D data analysis, the strength of our SNP-based genetic correlations and the gene-level overlap estimates decreased but remained statistically significant. The findings suggest that while there is a genetic association between T2D and GI disorders that partially reflects an underlying association with BMI, there is also a genetic overlap between the disorders not accounted for by BMI. Following adjustment for BMI in multivariable MR analyses, T2D remained causally associated with GERD, PUD and gastritis-duodenitis. While BMI is also associated with GI disorders in our study, the present results agree with findings for our genetic overlap assessment and reports in a recent MR study⁴ supporting a genetic association between T2D and GI disorders independent of BMI.” (Lines 584 – 593).

5. We believe that our response has addressed the reviewer’s concerns, particularly regarding how BMI could affect our results and conclusion. We note that the new addition did not alter our conclusion.

Comment 2: ad 2 and 4) My comment was directed at how the inclusion of the associations in the MHC region might have influenced the results of the correlation and pathway analyses. The authors now present the results of the correlation analyses excluding the MHC region in ST2. The results using the T2D GWAS of Mahajan et al. (2018) are identical to the results including the MHC region. This simply is not possible.

Response to comment 2

The point raised here by the reviewer was that it is impossible to obtain the same or closely related results in the analysis with and without the exclusion of the MHC region in the genetic correlation assessment between T2D (Mahajan et al.) and GI disorders using the LDSC method. The statement that *“this is simply not possible”* prompted us to conduct a thorough reassessment, mainly to rule out any instance of error or mistakes. In response, we took the following steps:

- (a) We performed another set of analyses starting from scratch and downloaded fresh data from the source to ensure no errors or mistakes. Links to the data are provided in Supplementary Table 1.
- (b) We excluded the MHC region chr6: 28477797 – 33448354 (GRCh37/hg19, reference for the MHC region from: <https://www.ncbi.nlm.nih.gov/grc/human/regions/MHC?asm=GRCh37>.) This region has been used in several studies, including PMID: 28449694⁵, PMID: 37644603⁶, and PMID: 33886574⁷.
- (c) We took each step carefully to ensure no error, mistake, or oversight. The coordinates excluded here were the same as in our first revision.
- (d) Given some authors often extend the MHC region to approximately 25–34 Mb (GRCh37/hg19)^{8,9}, we performed another set of analyses extending the excluded region to chr6: 25000000 to 34000000 (GRCh37/hg19).

- (e) Besides other assessments, for example, using different commands to test if the excluded MHC region remains in our data, we manually checked the LDSC ‘sumstats data’ (obtained after the munging step) to ensure the region was no longer present.
- (f) We then performed genetic correlation analyses using the newly prepared and validated data. We updated our methods section to reflect this new addition, which reads thus:

“To rule out the potential impact of the MHC region on our genetic correlation estimates, we performed further analyses by excluding the MHC region (chr6: 28477797 – 33448354, GRCh37/hg19) from our data. The MHC region used here is based on the reference from <https://www.ncbi.nlm.nih.gov/grc/human/regions/MHC?asm=GRCh37>, and it has been used in several studies⁵⁻⁷. Given some authors often extend the MHC region to approximately 25–34 Mb (GRCh37/hg19)^{8,9}, we performed another set of analyses extending the excluded region to chr6: 25000000 to 34000000 (GRCh37/hg19).” [Lines 750 – 755].

We obtained the same results reported in Supplementary Table 2, indicating no mistake or error in our analysis. Briefly, the result of our rerun analysis is shown below. We highlighted two lines in yellow to show that the results are the same, whether the MHC region is excluded from trait 1 only or both traits 1 and trait 2 (which is understandable).

Trait 1	Trait 2	Rg	se	Gcov int	Gcov int se	P
T2Dmahaj-No-MHC-1.txt.sumstats.gz	PUD (Wu et al. 2021)	0.29	0.04	0.02	0.01	2.03E-11
T2Dmahaj-No-MHC-1.txt.sumstats.gz	GERD (GERD: UKBB_QSKIN)	0.36	0.02	0.04	0.01	3.02E-59
T2Dmahaj-No-MHC-1.txt.sumstats.gz	Gastritis-duodenitis (PheCode 535)	0.32	0.04	0.03	0.01	2.75E-17
T2Dmahaj-No-MHC-1.txt.sumstats.gz	IBS-GWAS (Wu et al 2021)	0.13	0.04	0.00	0.01	1.24E-03
T2Dmahaj-No-MHC-1.txt.sumstats.gz	Diverticular disease (PheCode 562)	0.19	0.03	0.02	0.01	7.13E-12
T2Dmahaj-No-MHC-1.txt.sumstats.gz	IBD (Wu et al 2021)	0.06	0.04	0.00	0.01	1.10E-01
T2Dmahaj-No-MHC-1.txt.sumstats.gz	PUD (PheCode 531)	0.42	0.07	0.01	0.01	3.34E-10
T2Dmahaj-No-MHC-1.txt.sumstats.gz	GORD (Wu et al 2021)	0.31	0.03	0.01	0.01	1.12E-29
T2Dmahaj-No-MHC-1.txt.sumstats.gz	Lanzoprazole-GWAS-20003-UKB	0.38	0.05	0.01	0.01	3.16E-14
T2Dmahaj-No-MHC-1.txt.sumstats.gz	Gastritis-duodenitis (Watanabe et al)	0.23	0.06	0.01	0.01	4.08E-05
T2Dmahaj-No-MHC-1.txt.sumstats.gz	IBS (PheCode 564.1)	0.20	0.05	0.01	0.01	1.06E-04
T2Dmahaj-No-MHC-1.txt.sumstats.gz	Diverticular disease (K57-diagnosed)	0.12	0.03	0.00	0.01	1.79E-04
T2Dmahaj-No-MHC-1.txt.sumstats.gz	IBD (Watanabe et al)	-0.05	0.03	0.06	0.01	9.32E-02
T2Dmahaj-No-MHC-1.txt.sumstats.gz	GERD.UKB.Q-No-MHC.sumstats.gz	0.36	0.02	0.04	0.01	3.02E-59

Supported by our findings and available evidence, we believe that the issue raised has been addressed as follows:

- (i) The purpose of conducting the analysis with and without the MHC region is to assess whether the observed genetic correlation is influenced or biased by this region. It is important to note that LDSC incorporates a built-in mechanism to mitigate such bias. For instance, it was confirmed, while responding to a related question on the Google group for ‘ldsc users’ (https://groups.google.com/g/ldsc_users), that the program automatically excludes highly significant SNPs suspected to be outliers, including those from the MHC region, by setting a threshold for the maximum chi-square statistic.

‘...if LDSC detects highly significant SNPs it suspects are outliers (from the MHC region, for example) it will by default drop those top SNPs by imposing a threshold for maximum chi-square statistic’ (https://groups.google.com/g/ldsc_users/c/fEDtVcm5oc?pli=1).

This response aligns with our experience (especially in our past studies using LDSC). It suggests that the likelihood of bias from the MHC region impacting our analysis, even without a manual exclusion, is unlikely. The observation may explain why our analysis yields consistent results when evaluating the genetic correlation between T2D (Mahajan et al., 2018) and GI disorders, regardless of whether the MHC region is excluded. Moreover, the understanding from this response may underscore why many researchers often do not bother excluding this region when performing genetic correlation analysis using the LDSC approach.

- (ii) Concerning whether the MHC region could bias our study results or conclusions, the response in section (i) above has addressed this question in full. Additionally, we manually excluded the MHC region from our data, reran the LDSC analysis and obtained consistent results. Since the reviewer's concern was specific to the results for T2D Mahajan et al. data, it is essential to note that we utilised other T2D GWAS datasets, including from the United Kingdom Biobank and the Xue et al. 2018¹⁰, to test for reproducibility. Notably, we obtained consistent results across all these GWASs (Supplementary Table 2), further supporting our findings' reliability. Given the consistency in our results, using a range of data, we argue that there is no evidence that the MHC region biases the results or conclusions of our study.
- (iii) It is essential to note that the data analysed in our study are publicly available and easily accessible, and we have cited and provided links to the data sources both in-text and in Supplementary Table 1. The LDSC analysis method we employed is widely recognised and user-friendly, with documented scripts available via a link provided in our manuscript. Our specific approaches are documented in the method section, and we also included the coordinates of the MHC region excluded in our data, providing the opportunity to reproduce our findings and particularly assess whether the MHC region biases our findings or conclusion.
- (iv) Given the evidence we have provided here and in the first revision, we believe the concern raised here has been fully addressed.

Comment 3: With regard to the pathway analyses, the authors show, using IBD as example, that the exclusion of the MHC region (ST45) confirms their conclusion that immunological processes jointly underlie T2D and GI traits. This is not surprising, as IBD is an autoimmune disease. By looking at the pathway data for the other GI traits, the influence of the MHC region becomes obvious (ST40-44). I do not believe that immunological processes would be indicated in these traits if the MHC region would be removed from the analyses.

Response to comment 3

The reviewer's comment relates to the MHC region and whether its exclusion would alter the conclusion of our study regarding the immunological and inflammatory pathways and processes shared by T2D and GI disorders. In our first revision, we noted that excluding the MHC genes means pathways related to this region would not be enriched. Still, it does not necessarily rule out the involvement of the immune or inflammatory pathways in T2D

and GI disorders. We demonstrated this position using the genes overlapping T2D and inflammatory bowel disease (IBD).

To address the present comment, we conducted additional pathway-based functional enrichment analyses using:

- a. All the genes overlapping T2D and GI disorder at $P_{\text{gene}} < 0.05$ ($P_{\text{FCP}} <$). These genes are presented in Supplementary Tables 40, 42, 44, 46, 48, and 50. Moreover, our specific approaches to the pathway analysis are documented in the method section, thus providing an opportunity for replicating or confirming our findings.
- b. Genes overlapping T2D and GI disorders $P_{\text{gene}} < 0.05$ ($P_{\text{FCP}} <$) but excluding those in the MHC region (25–34 Mb, GRCh37/hg19). These genes are presented in Supplementary Tables 41, 43, 45, 47, 49, and 51, allowing the replication of our results.

Unlike in the first revision, where we only used genes overlapping T2D and IBD as an example to illustrate our position, we now performed the analysis for all the pairs of T2D and GI disorders. The results of these analyses are presented in Supplementary Tables 52 – 57.

Based on the re-analyses, our observations are as follows:

1. All MHC pathways were no longer enriched in the analysis excluding this region (which is understandable as the genes are no longer present). However, for some pairs of T2D and GI disorders, immunological or inflammatory pathways/processes were still overrepresented, aligning with our position in the first revision. Specifically:
 - (i) T2D – IBD:
Besides those that might play indirect roles, the following pathways are known for inflammatory or immunological function: ‘cytokine receptor binding’, ‘neurotrophin receptor binding’, ‘cellular response to peptide hormone stimulus’, ‘regulation of leukocyte differentiation’, ‘T cell differentiation’, ‘Th1 and Th2 cell differentiation’, ‘interleukin-11 signalling pathway’, ‘DNA damage response (only ATM dependent)’, ‘Notch signalling pathway’, and ‘mechanoregulation and pathology of YAP/TAZ via Hippo and non-Hippo mechanisms’.
 - (ii) T2D – Diverticular disease:
‘Transforming growth factor beta (TGF-beta) binding’ and ‘TGF-beta signalling pathway’ are known to have inflammatory or immune-related functions.
 - (iii) T2D – IBS:
Directly or indirectly, some of the identified pathways or processes have inflammatory or immunological functions, including: ‘thyroid hormone signalling pathway’, ‘Notch signalling pathway’, and ‘antiviral mechanism by IFN-stimulated genes’.
 - (iv) T2D – Gastritis-duodenitis:

‘Primary hypercortisolism’ may have potential implications for immune and inflammatory processes.

- (v) T2D – GERD: Enriched pathways or processes do not have direct immunological or inflammatory function. Nonetheless, cancer-related traits/processes enriched may involve these pathways.
- (vi) T2D – PUD: Pathways overrepresented do not have direct immunological or inflammatory roles. This finding is consistent with our position in the submitted manuscript.

We have updated our manuscript to reflect this new addition. Again, to maintain conciseness, we reported large results in the supplementary section and only summarised relevant results in-text. For example, in the results section, we added the following text:

“Given several pathways linked to the MHC region were among those overrepresented in this study, largely implicating pro-inflammatory and immunological-related mechanisms, we assessed whether excluding the region made any difference to our results or conclusion. Our findings suggest that excluding the MHC region does not necessarily rule out the involvement of the immune or inflammatory pathways among the biological mechanisms shared by T2D and GI disorders. For example, while all MHC pathways were no longer significantly enriched in the analysis excluding the region...” (Lines 500 – 516).

In the discussion section, added the following:

“Importantly, our findings suggest that excluding the MHC region does not necessarily rule out the involvement of immune or inflammatory pathways among the significantly enriched biological mechanisms shared by Type 2 Diabetes (T2D) and gastrointestinal (GI) disorders. For instance, while all MHC pathways were no longer enriched in the analysis excluding the region, inflammatory or immunological mechanisms or processes were still overrepresented for genes overlapping T2D and IBD, IBS, diverticular disease and gastritis-duodenitis.” (Lines 614 – 618).

2. Based on current findings, non-MHC inflammatory or immune-related pathways were enriched for other GI disorders, including IBS, diverticular disease, and Gastritis-duodenitis, not just IBD. In other words, the pathways were overrepresented for the GI disorders, except PUD and GERD. Even with the inclusion of the MHC region, we previously reported that pathways represented for PUD differ considerably from those of other GI disorders. Hence, current findings (the rerun analyses) align (primarily) with our position in the first revision. Therefore, excluding the MHC region does not alter our results or conclusion.
3. In our first revision, we noted that evidence supporting the MHC region in T2D and GI disorders was consistent across LAVA (local genetic correlation assessment) and the various levels of gene-based analyses. The question is, could the results regarding the MHC region bias our study or be false positive? We have responded to this question in the first revision and showed, using relevant evidence (for example, the LAVA software, which has built-in mechanisms for preventing false positives), that the occurrence of bias or false positives is improbable in our findings. Notably, MHC genes have been

implicated in some of these disorders, and we confirmed those in the present study (many of them were reported as sentinel genes, lines 376 – 385, and presented in Supplementary Tables 5 – 11), indicating that our findings related to this region are not strange or surprising. Furthermore, following an overlap assessment across the sentinel genes, we found some MHC genes shared across T2D and GI disorders (lines 386 – 396 and Supplementary Tables 12 – 15). Moreover, we achieved a similar result using the LAVA method and, subsequently, the pathway-based analysis, thus strengthening the evidence. Consequently, we firmly believe no justification exists for excluding the MHC region in our study.

4. Given the robust evidence in our study, we conclude as follows:

- Excluding the MHC region did not alter our overall results or conclusion.
- There are non-MHC genes identified in our research that are known for inflammatory and immune-related mechanisms.
- Given the consistent findings across different methods, the MHC results were not false positives, and there is no justification for removing MHC-related pathways or genes from our study.
- That consistent evidence supports the involvement of inflammatory or immunological mechanisms in our study, whether the MHC region was excluded or not; hence, our conclusion remains unchanged.

We believe that our detailed revisions and comprehensive responses, alongside the responses in our first revision, have addressed all the concerns raised by the reviewers.

Thank you for considering our revised manuscript, and we look forward to your response.

Yours Sincerely,

Dr Emmanuel Adewuyi

References

- 1 Niu, Y.-y., Aierken, A. & Feng, L. Unraveling the link between dietary factors and cardiovascular metabolic diseases: Insights from a two-sample Mendelian Randomization investigation. *Heart & Lung* **63**, 72-77 (2024).
- 2 Walker, V. M. *et al.* The consequences of adjustment, correction and selection in genome-wide association studies used for two-sample Mendelian randomization. *medRxiv*, 2020.2007.2013.20152413 (2020).
- 3 Hartwig, F. P., Tilling, K., Davey Smith, G., Lawlor, D. A. & Borges, M. C. Bias in two-sample Mendelian randomization when using heritable covariable-adjusted summary associations. *International Journal of Epidemiology* **50**, 1639-1650, doi:10.1093/ije/dyaa266 (2021).
- 4 Chen, J. *et al.* Gastrointestinal Consequences of Type 2 Diabetes Mellitus and Impaired Glycemic Homeostasis: A Mendelian Randomization Study. *Diabetes Care*, dc221385 (2023).
- 5 Matzaraki, V., Kumar, V., Wijmenga, C. & Zhernakova, A. The MHC locus and genetic susceptibility to autoimmune and infectious diseases. *Genome Biol* **18**, 76, doi:10.1186/s13059-017-1207-1 (2017).
- 6 Jiang, Y. *et al.* A genome-wide cross-trait analysis identifies genomic correlation, pleiotropic loci, and causal relationship between sex hormone-binding globulin and rheumatoid arthritis. *Hum Genomics* **17**, 81, doi:10.1186/s40246-023-00528-x (2023).
- 7 Chi, C. *et al.* Hypomethylation mediates genetic association with the major histocompatibility complex genes in Sjögren's syndrome. *PLoS One* **16**, e0248429, doi:10.1371/journal.pone.0248429 (2021).
- 8 Wu, Y. *et al.* GWAS of peptic ulcer disease implicates *Helicobacter pylori* infection, other gastrointestinal disorders and depression. *Nature Communications* **12**, 1146, doi:10.1038/s41467-021-21280-7 (2021).
- 9 Tesfaye, M. *et al.* Shared genetic architecture between irritable bowel syndrome and psychiatric disorders reveals molecular pathways of the gut-brain axis. *Genome Medicine* **15**, 60, doi:10.1186/s13073-023-01212-4 (2023).
- 10 Xue, A. *et al.* Genome-wide association analyses identify 143 risk variants and putative regulatory mechanisms for type 2 diabetes. *Nature Communications* **9**, 2941, doi:10.1038/s41467-018-04951-w (2018).

REVIEWERS' COMMENTS:

Reviewer #1 (Remarks to the Author):

The authors have answered the remaining questions, which has increased the value of their work.

Response to the editorial comments

Thank you for your message about our manuscript titled "Genome-wide cross-disease analyses highlight causality and shared biological pathways of type 2 diabetes with gastrointestinal disorders." We appreciate your efforts and those of the referee and editorial team in providing valuable feedback and facilitating the review process.

We have completed all the editorial formatting and editing requested and are now submitting this version for your formal acceptance. Briefly, we completed the following.

- We have revised our manuscript to meet format requirements and enhance accessibility.
- We have completed the editorial requests document (form) and will upload it as a related manuscript.
- We have completed and will upload the editorial policy checklist and the reporting summary form, as requested.

Thank you again for considering our work. We look forward to hearing back from you soon.

Best regards,

Dr Emmanuel ADEWUYI